# Safety and biologic activity of a bispecific T cell receptor targeting HIV Gag in males living with HIV: a first-in-human trial

Linos Vandekerckhove [1] ✉, Julie Fox [2], Borja Mora-Peris [3], Jordi Navarro [4,5], Sabine D. Allard[6], Alison J. Uriel[7], Santiago Moreno Guillén[8], Marta Boffito[9], Frank A. Post [10], Vicente Estrada[11], Beatriz Mothe [12,13,14], Mareva Delporte [1], Adel Benlahrech [15], Haseeb Rahman[15], James Clubley [15], Agatha Treveil [15], Jonathan Chamberlain[15], Rory Harrison[15], Miriam Hock [15], Yuan Yuan[16], Jason Wustner[16], Sylvie Moureau[15], Andrew D. Whale [15], Zoë Wallace[15], Praveen K. Singh[15], Kehmia Titanji[16], Lucy Dorrell[15] & Sarah Fidler [3]

HIV persistence in reservoirs despite antiretroviral therapy (ART) is a barrier to a permanent cure. We present the affinity-enhanced TCR bispecific IMC-M113V as a potential therapeutic for targeted HIV reservoir elimination. Preclinical studies demonstrate that IMC-M113V redirects T cells towards cells expressing the variable viral peptide, $Gag_{77-85}$, presented by HLA-A*02:01 at low copy number, without binding to HIV-negative cells. Here, we conduct a first-in-human, open-label single ascending dose study of IMC-M113V (1.6-15 μg) in twelve HLA-A*02:01-positive males living with HIV on suppressive ART (EudraCT number 2021-002008-11). Participants receive one intravenous infusion of IMC-M113V on Day 1 and are monitored through Day 29 to evaluate safety, tolerability (primary endpoints) and pharmacodynamic (PD) activity (secondary endpoint). IMC-M113V is well tolerated and not associated with any serious adverse event. PD activity is dose-dependent and strongest in participants with highly IMC-M113V-sensitive $Gag_{77-85}$ variant sequences. Thus, we provide a promising foundation to evaluate multiple and higher doses of IMC-M113V as a strategy for achieving ART-free virological control.

Durable antiretroviral (ART)-mediated suppression of HIV replication has led to dramatic reductions in mortality and normalisation of life-span for people living with HIV (PLWH). ART is not curative, however, and must be taken continuously for life due to the persistence of a reservoir of replication-competent HIV proviruses, which cause rebound viremia if treatment is interrupted[1]. Reservoirs are composed primarily of CD4 + T cells, but HIV DNA is also found in other cell types, including hematopoietic progenitors, macrophages, dendritic cells, microglial cells and astrocytes[2]. The complete eradication of reservoirs is an extremely ambitious goal that has to date only been achieved by

hematopoietic stem cell transplantation (HSCT) with CCR5Δ32/Δ32-deleted donor cells[3–7]. The high morbidity and mortality associated with HSCT restrict its application to people with life-threatening malignancies for which there are no other treatment options. Consequently, the intent of investigational therapies is to safely reduce reservoirs to levels that would lead to sustained virological control after ART withdrawal, i.e., a 'functional cure'[8].

Because the host immune response curtails HIV replication following acute infection and may, in rare cases, lead to a state of prolonged virological suppression in the absence of ART, novel modalities

that can recapitulate immune-mediated viral control and/or directly target reservoirs for elimination are under evaluation[9–12]. However, diverse immune evasion strategies used by HIV, including the acquisition of escape mutations that do not impede replication, could limit their effectiveness. For example, broadly neutralising antibodies (bNAb) in PLWH led to a delay in viral rebound for up to 6 months after ART interruption, and for even longer durations in PLWH who initiated ART very early, but typically in individuals who had not acquired detectable bNAb resistance mutations prior to treatment[13–16]. Furthermore, this post-treatment control was largely dependent on maintaining plasma bNAb levels above a predefined threshold, although in some cases modest reductions in viral reservoirs were observed[13,15,16]. Other technologies, such as adoptive cell therapies, bispecific antibodies and gene editing, have shown promise in animal models, and clinical proof of concept is beginning to emerge[17–20]. Still, a broadly applicable and scalable therapy that fulfils the target product profile of a functional cure remains elusive[21].

Immune mobilising monoclonal T cell receptors against cancer (ImmTAC) and viruses (ImmTAV) are soluble bispecific proteins comprised of an affinity-enhanced T cell receptor fused to a CD3-specific single chain variable fragment (scFv) (Fig. 1a). The TCR binds to a specific peptide when presented by HLA molecules on the target cell surface; the scFv binds to CD3 on any T cell in the vicinity, leading to the formation of an immune synapse and elimination of the target cell[22,23]. The ImmTAC platform has been clinically validated by tebentafusp, which binds to an HLA-A*02:01-presented peptide derived from the melanocyte antigen, gp100, and which conferred a significant survival benefit to HLA-A*02:01-positive patients with metastatic uveal melanoma in a randomised controlled trial[24]. ImmTAV molecules are an adaptation of this technology and aim to provide targeted

elimination of viral reservoirs in people with chronic hepatitis B[25], HIV and other persistent infections[26]. We previously reported that an ImmTAV molecule that specifically recognises an immunodominant HLA-A*02:01-restricted HIV peptide, Gag77–85, along with its common escape variants, could redirect T cells to eliminate HIV-infected cells from ART-suppressed PLWH after viral reactivation in vitro[27]. This ImmTAV molecule was active at picomolar concentrations, indicating exquisite sensitivity to the Gag peptide, which may be crucial to its efficacy in vivo, given that HIV proteins are expressed at very low levels in reservoirs[28–30]. Further engineering was undertaken to produce the clinical candidate drug IMC-M113V, which was advanced to a first-in-human (FIH) study in HLA-A*02:01-positive PLWH on a suppressive ART regimen.

Given the excellent overall safety profile of current ART and the benefit/risk considerations for a therapy with a new mechanism of action against HIV, IMC-M113V was initially evaluated in a single ascending dose (SAD) study to identify tolerable and biologically active doses. To ensure safety, principal eligibility criteria were: documented virological suppression on ART for at least one year, current CD4 + T cell count >500 cells/µL and CD4 + T cell nadir >200 cells/µL. In addition, the duration of ART at study entry was restricted to ≤7 years to maximise the potential to detect pharmacodynamic (PD) activity against Gag-expressing cells, which are indicative of a transcriptionally and translationally active HIV reservoir and which decline in frequency with long-term viral suppression[31,32]. Here, we present both the pre-clinical data supporting the clinical study rationale, and the results of the SAD study, which demonstrate the safety, tolerability and initial PD activity of IMC-M113V. These data inform the evaluation of IMC-M113V in multiple dose schedules, where it has the potential to confer post-treatment viral control through targeted reservoir reduction.

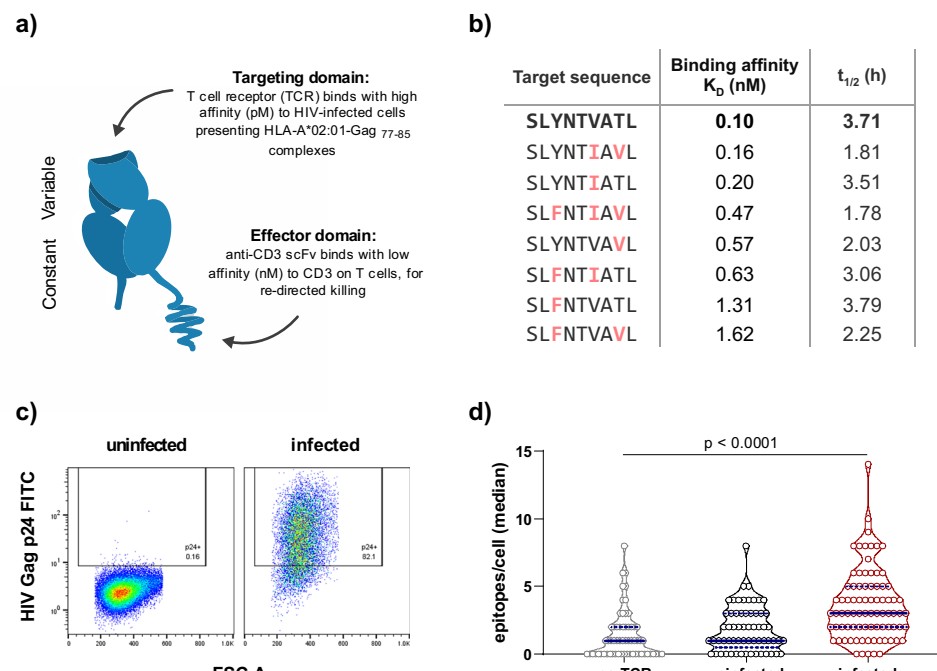

**Fig. 1 | IMC-M113V is active against prevalent variants of Gag77–85. a** Schematic of IMC-M113V consisting of an affinity-enhanced TCR as the targeting domain and an anti-CD3 single-chain variable fragment (scFV) as the effector domain. **b** The binding affinity of IMC-M113V to the Gag77–85 cognate peptide sequence and the 8 most prevalent variants was determined by surface plasmon resonance (25 °C) and is presented as $K_D$ values. Stability of the Gag77–85 variants in complex with HLA-A*02:01 was determined from $t_{1/2}$ at 37 °C. **c** Infection of HLA-A*02:01-transduced C8166 cells with HIV-1 IIIB was confirmed by intracellular HIV Gag p24 staining.

**d** Confirmed uninfected or infected C8166 cells were stained with the rabbit Fc-tagged TCR domain of IMC-M113V^RES and goat anti-rabbit CF640R. TCRs bound to peptide-HLA complexes on infected cells were quantified by total internal reflection microscopy. Each dot represents one cell. No TCR control represents infected cells stained with goat anti-rabbit CF640R only. Horizontal bars indicate median values, and indigo dotted lines indicate upper and lower quartiles. Statistical significance was determined using 1-way analysis of variance (ANOVA) with Tukey multiple comparisons test.

## Results

### Binding kinetics and activity of IMC-M113V against prevalent Gag$_{77-85}$ variants

IMC-M113V was derived from a previously described ImmTAV molecule (m121) that was modified to meet manufacturing specifications[27]. The TCR domain was engineered to bind HLA-A*02:01-Gag$_{77-85}$ (cognate sequence, SLYNTVATL) complexes with an enhanced affinity of ~100 pM ($K_D$ at 25 °C) (Fig. 1a, b). Some of the preclinical assays described here were performed with the closely related precursor molecule, IMC-M113V$^{RES}$, which differs from IMC-M113V by a single amino acid and demonstrates a similarly enhanced binding affinity to the cognate peptide HLA (pHLA). This residue is located within the TCR domain and was mutated to support the manufacturability of IMC-M113V by reducing the risk of a non-enzymatic post-translational modification.

Gag$_{77-85}$ variants have arisen over the course of the HIV pandemic because of immune selective pressure and/or recombination events, with Y79F representing the most prevalent variant globally (Supplementary Fig. 1a)[33]. To address anticipated inter- and intra-individual viral diversity, the TCR domain of IMC-M113V was engineered to achieve >1000-fold enhancement in affinity for the 8 most prevalent (global frequency ≥ 2%) Gag$_{77-85}$ variants identified using the Los Alamos National Laboratory HIV sequence database (Supplementary Fig. 1a), all of which are stably presented by HLA-A*02:01 (Fig. 1b). TCR binding affinities for the variants ranged from 0.16 to 1.62 nM (Fig. 1b).

The potency of IMC-M113V against T2 cells presenting the Gag$_{77-85}$ cognate and variant peptides was assessed in a T cell redirection assay with titrated peptide concentrations. Peptide half-maximal effective concentration (EC$_{50}$) values correlated with the measured affinities and ranged from 0.31 nM for the V82I variant to 5.29 nM for the Y79F / T84V double mutant across the two effector donors tested (Supplementary Fig. 1b). To confirm binding of the TCR to Gag-positive cells, the TCR domain (Fc-tagged and lacking the anti-CD3 scFv) of IMC-M113V$^{RES}$ was used to stain an HLA-A*02:01-transduced HIV-infected CD4 + T cell line (C8166) (Fig. 1c), which was then analysed using total internal reflection microscopy. This enabled quantification of TCR-bound pHLA complexes (epitopes) on the cell surface[23,34]. HLA-A*02:01-Gag$_{77-85}$ complexes were detected at a median copy number of 3/cell (range 0–14) on infected cells. Despite some background signal (range 0–8) on uninfected cells due to non-specific binding of the secondary antibody, pHLA density was significantly higher on infected cells ($p$ < 0.0001 (Fig. 1d). In a parallel experiment to determine the HLA-A*02:01-Gag$_{77-85}$ density on T2 cells pulsed with titrated peptide concentrations, 1 nM peptide yielded a median epitope count of 2 (range 0–13) pHLA/cell (Supplementary Fig. 1c). Taken together, these results indicate that IMC-M113V is sensitive to very low levels of viral pHLA complexes expressed on the cell surface.

### IMC-M113V potency against HIV-infected cells in vitro

The potency of IMC-M113V against HIV-infected cells was determined using an infected cell elimination assay, in which HLA-A*02:01-transduced HIV-infected C8166 cells were co-cultured with purified CD8 + T cells from healthy donors ($n$ = 6) for 7 days, together with titrated concentrations of IMC-M113V. Dose-dependent elimination (calculated using the equation described in "Methods") was observed across all effector donors, with a mean EC$_{50}$ value of 6.5 pM (range 1.2–13.1 pM, Fig. 2a). IMC-M113V also eliminated targets infected with the most prevalent global variant, Y79F, in a dose-dependent manner ($n$ = 4 effector donors, mean EC$_{50}$ value < 100 pM), albeit less efficiently than the cognate virus, consistent with the weaker binding affinity of IMC-M113V for the Y79F variant (Figs. 1b and 2b and Supplementary Fig. 1b).

### In vitro assessment of IMC-M113V off-target reactivity

To assess the potential for off-target reactivity, IMC-M113V was tested in clinically validated screening assays incorporating whole blood and a panel of normal cells associated with vital organs obtained from HLA-A*02:01-positive healthy donors[35]. In these assays, interferon-γ (IFN-γ) release, alone or together with other cytokines, was used as a sensitive readout for T cell activation in vitro (Fig. 2c). No statistically significant cytokine production above background levels was observed in whole blood, or in co-cultures of immune cells and cardiomyocytes, astrocytes or lung epithelial cells, when incubated with IMC-M113V at concentrations below 10 nM (Fig. 2c). As the biologically active dosing range was anticipated to be 0.1–1 nM, the reactivity at 10 nM was not considered to be a concern.

### FIH safety of IMC-M113V in people living with HIV

Given the HLA restriction of IMC-M113V, the study began with pre-screening of potentially eligible participants ($n$ = 89) for the HLA-A*02:01 allele. We estimated the prevalence of HLA-A*02:01 to be in the range of 23–45% among PLWH populations in the countries where the study was conducted, and this was borne out in the pre-screening data (Fig. 3)[36,37]. Of the 33 HLA-A*02:01-positive participants who were eligible for screening, 16 completed screening, of whom four were excluded due to a CD4 + T cell count <500 cells/μL. The first and last participants were enrolled on 6 July 2022 and 12 December 2022, respectively, at 7 hospital sites in the United Kingdom, Belgium and Spain. Altogether, 12 participants were enrolled in three single ascending dose cohorts (Fig. 3). Although females were eligible and invited to participate, all study participants were male, with a median age of 38 years (range 27–60 years). The median duration of ART on Day 1 was 3.5 years (range 2–7 years), the median CD4 + T cell count was 882 cells/μL (range 466–1460 cells/μL), and the median CD4 + T cell nadir was 550 cells/μL (range 330–846 cells/μL) (Table 1).

Participants were enrolled in accordance with the planned dose escalation scheme (described in "Methods"), with a single individual being enroled at each of the first two dose levels (1.6 μg and 5 μg), to minimise exposure to potentially sub-therapeutic doses[38]. At the 15 μg dose level, 4 participants were enroled initially, of whom 2 met the pre-specified criteria for PD activity. This cohort was subsequently expanded to a total of 10 participants to enable further evaluation of this PD activity (Fig. 3). As the primary endpoint, safety and tolerability of single doses of IMC-M113V were assessed by incidence and severity of treatment-emergent adverse events (TEAE). Fifteen TEAEs were recorded in 6/12 (50%) participants: 7 were deemed unrelated to the study drug, and 8 were considered possibly related (Supplementary Table 1). Of the six participants who experienced TEAEs, 5 (83%) experienced the following possibly related events: fatigue (1 Grade 1, 1 Grade 2) lasting 2–4 days, phlebitis, herpes simplex, itch, skin redness (all Grade 1) and two episodes of eczema (1 Grade 1, 1 Grade 2). Five of the 6 aforementioned skin AEs were observed in the same participant. Apart from one procedure-related event, the onset of all these AEs was 5–20 days after administration of IMC-M113V. No participant developed fever, cytokine release syndrome or neurotoxicity of any grade, nor were there any serious adverse events. There were no laboratory AEs of Grade ≥ 3 severity. CD4 + T cell counts were >500 cells/μL post-dosing in all participants, including those who had a pre-dose (Day 1) CD4 + T cell count of <500 cells/μL ($n$ = 2) despite meeting all eligibility criteria at screening. All plasma HIV RNA (viral load, pVL) values were below the limit of detection or detected at ≤ 40 copies/mL post-dosing and through follow-up.

### Pharmacokinetics of IMC-M113V

Serum IMC-M113V concentrations measured pre-dose and up to 24 h after IV administration were used to derive pharmacokinetic profiles for each dose level using non-compartmental analysis. IMC-M113V was detectable at all dose levels administered, and peak concentrations were observed immediately after infusion (Fig. 4). Overall, exposure increased in a dose-proportional manner, with C$_{max}$ values ranging

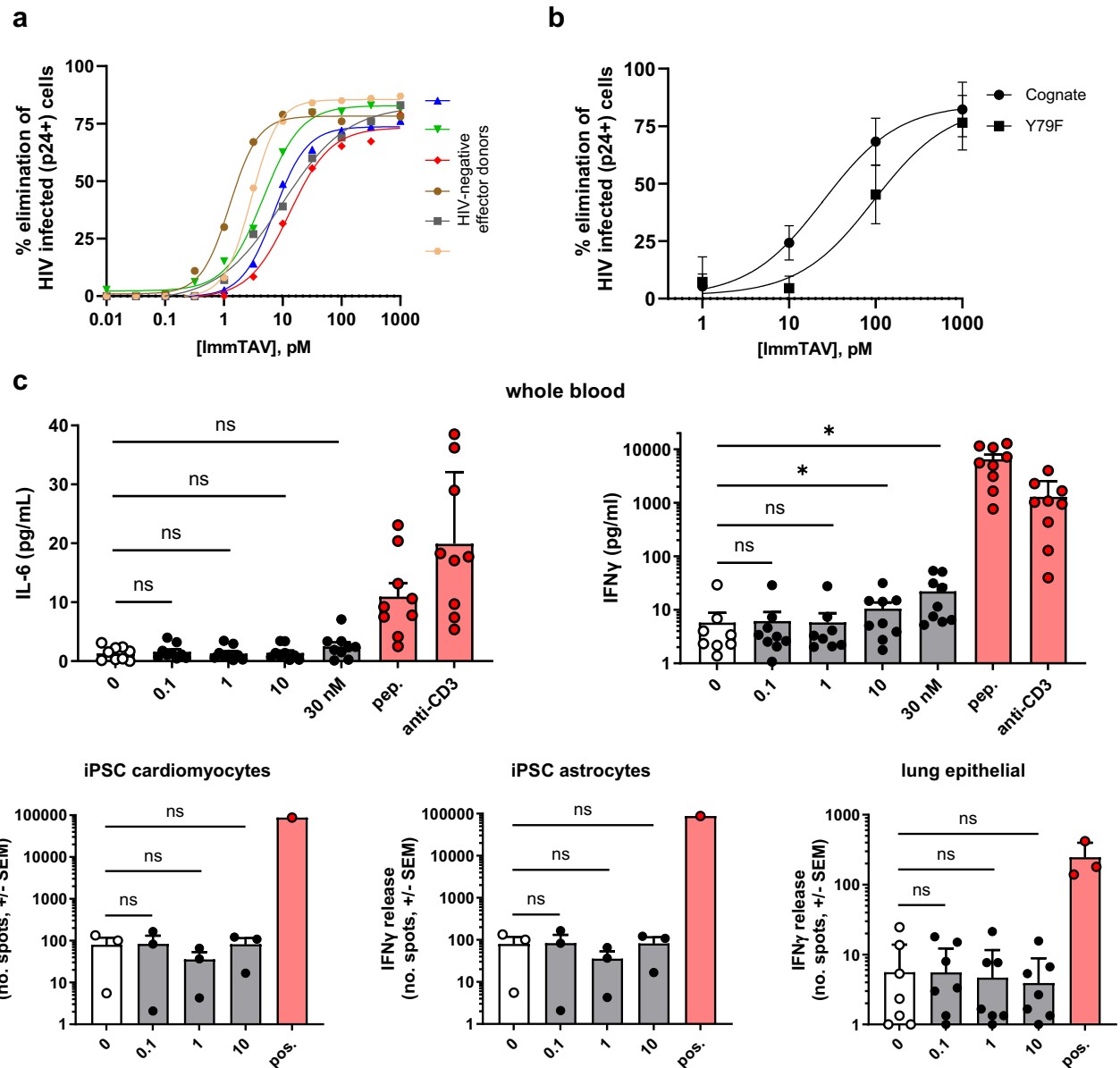

**Fig. 2 | IMC-M113V eliminates HIV-infected cells in vitro with picomolar potency and high specificity. a** HLA-A*02:01-transduced HIV-1 IIIB-infected C8166 cells were co-cultured with purified CD8 + T cells from HIV-negative donors ($n = 6$) at a ratio of 2:1 in the presence of increasing concentrations of IMC-M113V for 7 days. Percentage elimination of infected cells was derived from the reduction in Gag p24+ cells in the presence of IMC-M113V, as determined by intracellular staining. **b** Potency of IMC-M113V against the most prevalent Gag$_{77-85}$ variant, Y79F, was assessed in the infected cell elimination assay as described in (**a**), using mutated pNL4-3 viruses expressing either the Gag$_{77-85}$ cognate or Y79F sequence. Data indicate mean ± SD infected cell elimination values obtained with purified CD8 + T cell effectors from 4 HIV-negative donors. **c** Potential cross-reactivity of IMC-M113V with normal tissues was assessed in assays with fresh whole blood (top panels) or induced pluripotent stem cell (iPSC)-derived cardiomyocytes, iPSC-

derived astrocytes, and lung epithelial cells (bottom panels) from HLA-A*02:01-positive healthy donors. Cells were incubated with IMC-M113V (0.1-30 nM), medium alone (0, no drug control) or IMC-M113V (5 nM, whole blood assay; 1.1 nM, normal cells and iPSC) and Gag$_{77-85}$ peptide (pep. or pos., 10 μM, positive control). Stimulatory anti-CD3 antibody was included as an additional positive control for the whole blood assay. Cytokine release was determined after 4 h (whole blood) or 24-48 h (lung epithelial cells or iPSC) by Meso Scale Discovery (MSD) immunoassay or interferon-gamma enzyme-linked immunospot (IFN-γ ELISpot). Datapoints represent mean ± SD of 3 replicates for individual whole blood donors ($n = 9$) or unique combinations of the normal cell lots and effector donors tested ($n = 9$ lung epithelial cells and iPSC cardiomyocytes; $n = 3$ iPSC astrocytes). Statistical significance was determined using two-sided Wilcoxon matched-pairs signed rank and Holm-Šídák multiple comparison tests. ns = not significant; *$p = <0.05$.

from 171 to 3,650 pg/mL and area under the curve (AUC$_{last}$) values from 1220 to 45,400 h*pg/mL. Serum IMC-M113V levels decreased below the lower limit of quantification (LLOQ) by 12 h post-administration of the 1.6 μg dose but remained detectable at all time-points up to 24 h for the 5 and 15 μg doses. Based on the 15 μg dose ($n = 10$), clearance (CL) ranged from 2.03 to 12.8 L/day; the steady-state volume of distribution (Vss) was 1.93–11.2 L, and the terminal half-life was estimated to be 14.5–21.6 h.

## Pharmacodynamic activity of IMC-M113V

A biomarker-guided dose escalation strategy was implemented in this SAD study in order to identify a tolerable dose range on which to base a future multiple dose schedule. Target-specific and dose-proportional increases in proinflammatory cytokines and chemokines in the serum have been observed following treatment with tebentafusp, consistent with its mechanism of action as a T cell engager[39,40]. As shown in the preclinical studies described earlier, IFN-γ release is a direct and

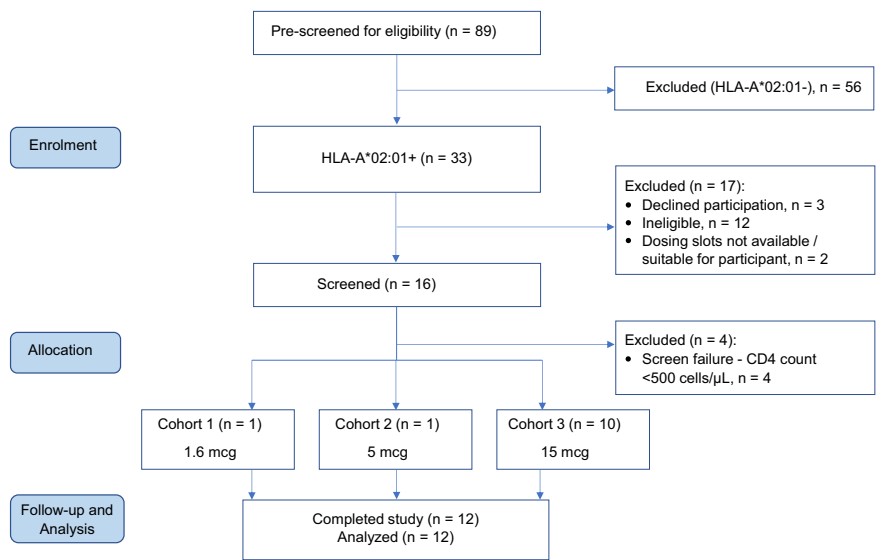

**Fig. 3 | Soluble T cell receptors in virus eradication (STRIVE) study.** CONSORT flow diagram showing identification, screening and disposition of study participants.

## Table 1 | Participant demographics and baseline clinical characteristics

| ID | Race/ethnicity | IMC-M113V dose (μg) | Years HIV + | Years on ART | Pre-ART CD4 + T cell nadir (cells/μL) | CD4 + T cell count on Day 1[b] (cells/μL) | Viral subtype |
|---|---|---|---|---|---|---|---|
| P01 | N. African, Not Hispanic/Latino | 1.6 | 3 | 3 | 395 | 927 | B |
| P02 | White | 5 | 6 | 5 | 668 | 1239 | Unknown |
| P1 | White | 15 | 3 | 3 | 700 | 1041 | Unknown |
| P2 | Black | 15 | 28[a] | 4 | 580 | 474 | D |
| P3 | Hispanic/Latino | 15 | 4 | 2 | 570 | 730 | Unknown |
| P4 | Hispanic/Latino | 15 | 4 | 4 | 333 | 466 | B |
| P5 | Unknown | 15 | 3 | 3 | 846 | 1460 | A6 |
| P6 | White | 15 | 6 | 6 | 599 | 534 | G |
| P7 | White | 15 | 12 | 7 | 530 | 1301 | B |
| P8 | White | 15 | 2 | 2 | 330 | 1022 | BC recombinant |
| P9 | White | 15 | 11 | 6 | 441 | 622 | Unknown |
| P10 | White | 15 | 6 | 2 | 466 | 836 | CRF50_A1D (5.3%) |

[a]Previous HIV long-term non-progressor, initiated ART after viral load increase to 13,000 copies/mL and CD4+ cell count decline to the nadir value indicated. His CD4 cell count at Screening was 760 cells/μL.
[b]Pre-dose.

sensitive measure of T cell activation. However, interleukin-6 (IL-6)-producing cells, which are largely of myeloid and stromal origin[41,42], can amplify the initial T cell activation signal, potentially offering higher sensitivity to detect target engagement[39]. To determine whether a robust IL-6 signal could be detected in the context of low Gag$_{77-85}$ peptide presentation, which is anticipated in ART-treated PLWH, we first assessed IMC-M113V$^{RES}$-mediated IL-6 induction in a T cell activation assay with titrated peptide concentrations. We confirmed that IL-6 production by peripheral blood mononuclear cells (PBMC) occurred only in the context of TCR engagement with HLA-A*02:01-Gag$_{77-85}$, at peptide concentrations as low as 100 pM (Supplementary Fig. 2 and Fig. 2C).

In the clinical study, serum IL-6 was quantified pre-dose, 8 and 24 h post-dose, and on Day 8, together with IFN-γ, TNF, IL-2, IL-8, IP-10, and IL-10. Baseline IL-6 values were below the LLOQ in 9/12 participants, while the remaining three participants all had values within the normal range. A small increase of 2-3-fold from baseline in IL-6 was noted in each of the single participant cohorts, with a peak occurring at 8 h for the 1.6 μg dose and 24 h for the 5 μg dose. Five of the 10 participants in the 15 μg dose cohort showed a > 4-fold increase in IL-6 from baseline, of whom two had a > 10-fold rise (Fig. 5a). The highest recorded value in any participant was 62.8 pg/mL (Fig. 5 and Supplementary Fig. 3). By Day 8, IL-6 levels had declined to baseline or below 10 pg/mL. Modest changes were observed in other cytokines in the 15 μg dose cohort, with 1/10 and 3/10 participants showing a > 2-fold rise in IFN-γ and IL-8, respectively, at 8–24 h post-dose, and 1/10 showing a nearly 4-fold rise in TNF on Day 8 (Supplementary Fig. 3). We also assessed T cell activation ex vivo in a subset of participants in the 15 μg dose cohort with available PBMC samples, pre- and post-dose, by flow cytometry (Supplementary Figs. 4 and 5). This showed a post-dose increase in the frequency of CD4+ and CD8 + T cells expressing the early activation marker, CD69, in 3/6 participants, together with a significant increase in CD69 mean fluorescence in both cell subsets overall (n = 6, Fig. 5d). In addition, after ex vivo stimulation with phorbol myristate acetate (PMA)/ionomycin, IFN-γ, granzyme B (GzB), and perforin expression were increased post-dose in several participants, with significant changes being detected in CD8 + T cells for GzB and perforin (n = 5, Fig. 5d).

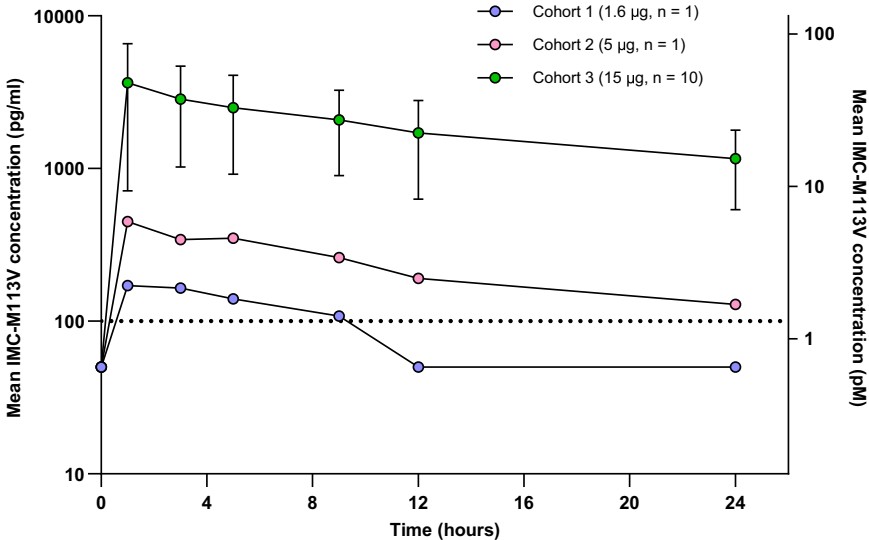

**Fig. 4 | Pharmacokinetic profile of single doses of IMC-M113V.** Serum concentrations of IMC-M113V were measured pre-infusion, at the end of infusion and at 2, 4, 8, 12 and 24 h post-end of infusion. Drug concentrations for all dose cohorts (mean ± SD for the 15 µg dose cohort) are indicated.

To explore the relationship between the IL-6 response and baseline HLA-A*02:01-Gag$_{77-85}$ expression, we quantified HIV cell-associated RNA (HIV CA-RNA) as a measure of the active reservoir. We also sequenced *gag* transcripts to determine the presence of Gag$_{77-85}$ cognate or variant sequences, given that IMC-M113V binding affinity for pHLA is affected by known mutations in this epitope (Fig. 1b and Supplementary Fig. 1b)[43]. Median HIV CA-RNA copy number was 639 (range 112–2777) copies / 10$^6$ TBP (TATA-box binding protein) for the 15 µg cohort. Gag$_{77-85}$ sequences detected in HIV CA-RNA included previously characterised variants (Fig. 1b and Supplementary Fig. 1) and some less prevalent variants for which the IMC-M113V$^{RES}$ binding affinity was determined reactively (Fig. 5b and Supplementary Fig. 6). The relationship between baseline HIV CA-RNA copy number, dominant Gag$_{77-85}$ sequence, and peak IL-6 level post-dose is shown in Fig. 5c. IL-6 elevations of >10-fold were observed in 2/3 of the participants with picomolar affinity Gag$_{77-85}$ variants; the measured affinities to these variants ranged from 76 to 178 pM. One participant with a 178 pM affinity variant (cognate sequence) but no IL-6 response had very low HIV CA-RNA levels (112 copies/10$^6$ TBP). In contrast, IL-6 elevations of >10-fold were observed in 0/7 participants with nanomolar affinity Gag$_{77-85}$ variants, for which the measured affinities ranged from -1.3 to 261 nM. Although the study was not powered to test the correlation between the IL-6 response and HIV CA-RNA expression, a trend was observed for the three participants with a picomolar Gag$_{77-85}$ variant (Pearson $r^2 = 0.87$) but not for the seven participants with nanomolar variants (Pearson $r^2 = 0.02$). These observations are consistent with in vitro data showing that the magnitude of IMC-M113V-mediated cytokine response was dependent on both the peptide concentration and the affinity of IMC-M113V for the peptide variant (Supplementary Figs. 1b and 2).

### Effect of a single 15 µg dose of IMC-M113V on the HIV reservoir

HIV CA-RNA and intact proviral DNA were quantified pre (Day 1) and post dose (Days 8 and 29) in an exploratory analysis to determine any impact on the size of the reservoir. Values did not change significantly throughout follow-up (Fig. 6).

### Discussion

We report on the preclinical and initial clinical evaluation of IMC-M113V, an affinity-enhanced soluble TCR bispecific molecule that targets a well-described HLA-A*02:01-presented HIV Gag peptide. IMC-M113V was specifically developed to overcome key mechanisms that

enable HIV to evade host defences. First, the known sequence variability within circulating HIV isolates, which could hinder cytolysis of cells bearing mutated Gag$_{77-85}$ sequences, was addressed in the molecule design, which started with a wild-type TCR that was inherently capable of recognising multiple viral variants[27,44]. This polyspecificity was maintained throughout the TCR affinity enhancement using phage display. Activity against multiple prevalent variants of the Gag$_{77-85}$ epitope, as identified in the Los Alamos HIV sequence database, was then confirmed in T cell redirection assays, with the observed variation in potency correlating with the differential binding affinities of the TCR domain for these variants. Second, engineering of the TCR domain of IMC-M113V to picomolar affinity was a crucial step towards ensuring it had the capacity to eliminate HIV-infected cells presenting viral peptides at very low levels (< 10 pHLA complexes/cell). This was confirmed in infected cell elimination assays, with a Nef-competent viral isolate presenting the cognate peptide sequence, demonstrating that Nef-mediated downregulation of HLA class I did not impair recognition of the Gag$_{77-85}$ epitope[30]. Overall, the preclinical data were consistent with previous observations made in an ex vivo study with a precursor molecule to IMC-M113V[27]. Third, we addressed the risk of new escape variants emerging during treatment with IMC-M113V in the study design, by administering IMC-M113V to PLWH who were virologically suppressed on ART[45].

The primary objective of the clinical study was to assess the safety and tolerability of IMC-M113V. Single doses ranging from 1.6-15 µg were well tolerated by PLWH on suppressive ART regimens with normal CD4 + T cell counts and were not associated with cytokine-release syndrome (CRS), nor any serious or severe adverse events. The absence of CRS contrasts with our clinical experience with ImmTAC molecules in solid tumours and with published reports on treatment with other T cell-engaging therapies in haematological malignancies, for example, blinatumomab and tisagenlecleucel[46,47]. The incidence and severity of CRS in oncology settings are thought to be related to both the initial drug dose and the tumour mass (on-target toxicity). Therefore, we speculate that the absence of CRS in this study, which was consistent with the generally low levels of all serum cytokines post-dosing, is indicative of the low doses of IMC-M113V tested to date, and/or the anticipated low frequency of Gag-positive cells within blood and lymphoid tissue in the study participants, based on published analyses of HIV reservoir composition in PLWH on ART[28,40,41]. While increasing the dose of IMC-M113V may increase the risk of CRS, experience with ImmTAC molecules and other CD3 bispecific biologics has shown that

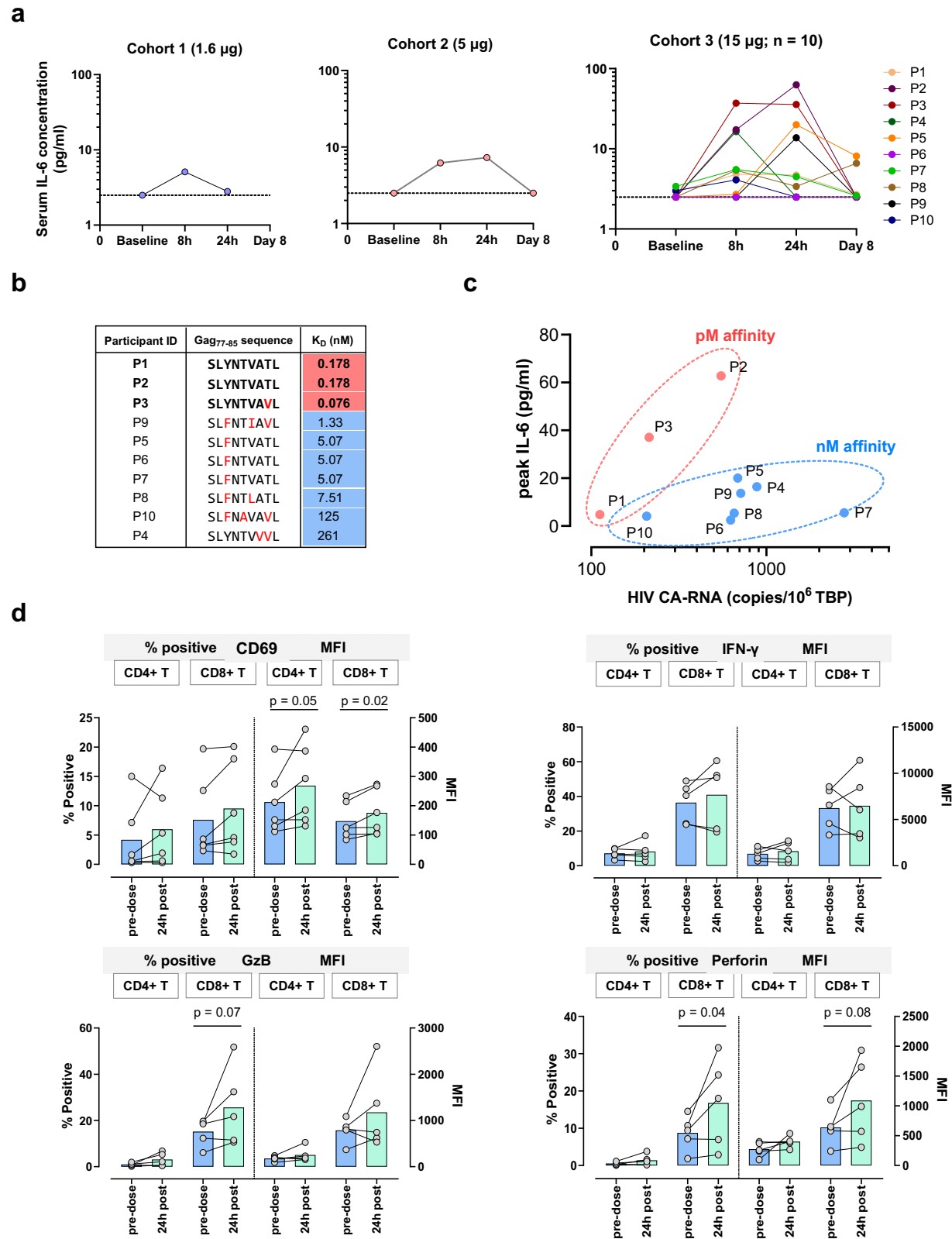

step dosing can induce tachyphylaxis, thereby enabling safe exposure to the higher dose levels that may be required for clinical efficacy[48,49]. As the 15 μg dose tested in this study was well tolerated, it could be used as an initial step dose in a future MAD schedule.

Secondary objectives of the study were to characterise the PK profile of IMC-M113V and to determine changes in the systemic immune response, including serum cytokines, as indicators of PD

activity. Serum IL-6 was used in this study to guide dose escalation decisions, not only because of its role in CRS pathogenesis, but because it appears to be a highly sensitive biomarker of target engagement by ImmTAX molecules, emerging earlier and with greater amplitude than T cell-derived cytokines[39]. Additionally, the criterion of a > 4-fold increase in serum IL-6 was informed by observations with tebentafusp in patients with metastatic uveal melanoma that showed a

**Fig. 5 | Identification of biologically active doses of IMC-M113V. a** Serum IL-6 was quantified in samples obtained pre-infusion, 8 and 24 h and 7 days post-infusion. Each line represents an individual study participant. The lower limit of quantification (2.5 pg/mL) is indicated by a dotted line. **b** IMC-M113V$^{RES}$ binding affinity (37 °C) was determined for Gag$_{77-85}$ variants identified in pre-dose HIV cell-associated RNA (CA-RNA) extracted from participants' peripheral blood CD4+ cells (15 μg cohort). The dominant sequence for each participant is shown. Relative frequencies of identified variants are shown in Supplementary Fig. 6. **c** Peak serum IL-6 response was plotted against pre-dose HIV CA-RNA value. Participants with picomolar and nanomolar affinity viral variants are indicated in pink and blue, respectively.

**d** Expression of T cell activation and cytolytic molecules pre- and 24 h post-dosing with IMC-M113V (15 μg): PBMCs collected pre-dose (blue bars) and 24 h post-dose (green bars) were stained for surface markers (*n* = 6) or stimulated ex vivo with PMA and ionomycin in the presence of brefeldin A (*n* = 5). Percentage of positively stained cells (left y axes) and mean fluorescence intensities (MFI, right y axes) are shown for CD69 (top left), interferon-gamma (IFN-γ, top right), granzyme B (GzB, bottom left) and perforin (bottom right). Circles represent percentage positivity or MFI of individual participants, whilst bars represent mean values. A two-sided paired *t*-test was used to compare pre-dose and post dose groups; *p* values < 0.05 were considered statistically significant.

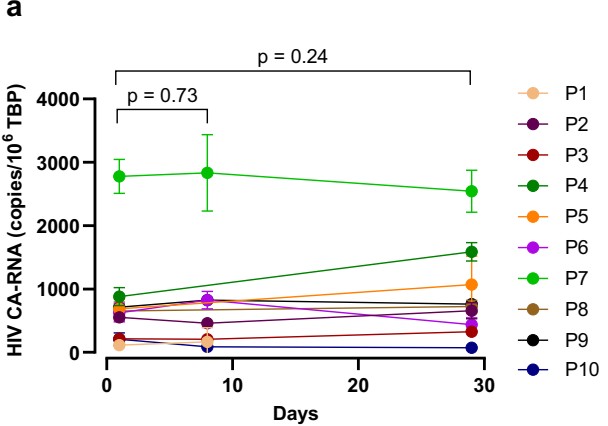

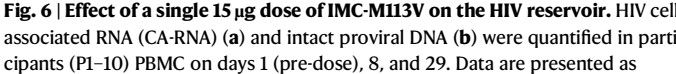

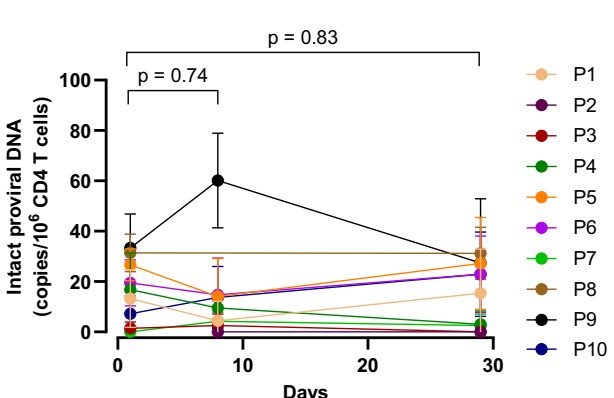

**Fig. 6 | Effect of a single 15 μg dose of IMC-M113V on the HIV reservoir.** HIV cell-associated RNA (CA-RNA) (**a**) and intact proviral DNA (**b**) were quantified in participants (P1–10) PBMC on days 1 (pre-dose), 8, and 29. Data are presented as mean ± SD from a minimum of two replicates. *p* values were calculated using a mixed-effects model and Šídák multiple comparison test.

strong association with overall survival, suggesting that it could serve as a surrogate for a durably effective immune response[39,41]. In the present study, we observed dose-dependent elevations in serum IL-6 following IMC-M113V administration, with kinetics consistent with the PK profile. Furthermore, the strongest IL-6 responses were detected in participants with highly drug-sensitive virus, as determined by drug-target binding affinity in vitro, and with HIV *gag* CA-RNA copies >200 copies/10⁶ TBP in peripheral blood, indicative of a substantial transcriptionally active reservoir. Other proinflammatory cytokines were minimally increased. However, upregulation of CD69 in CD4+ and/or CD8 + T cells was noted in some of the participants, together with increased expression of IFN-γ, GzB and perforin after activation in vitro, indicating that effector function was enhanced by IMC-M113V. These PD effects are commensurate with the expected low expression of HIV Gag in viral reservoirs and the low doses of IMC-M113V evaluated in this study. Similar observations have been reported following treatment of minimal residual disease-positive B-lineage acute lymphoblastic leukaemia with the bispecific T cell engager blinatumomab, where some patients showed an IL-6 response in the absence of detectable IFN-γ[50]. Taken together, these data support the proposed mechanism of action, i.e., IMC-M113V-mediated T cell activation upon specific engagement with pHLA complexes on Gag-expressing cells, with IL-6 secretion by myeloid and/or stromal cells occurring as a downstream effect.

To confirm the clinical relevance of the specificity profile of IMC-M113V in PLWH, we determined the IMC-M113V binding affinity for Gag$_{77-85}$ variants identified from bulk sequencing of HIV CA-RNA in the SAD study participants at baseline. This approach identified the dominant species in all participants, together with some low-frequency minority target variants in several individuals. In addition to detecting previously characterised variants, we identified 3 new variants, to which IMC-M113V bound with nanomolar affinity. We

predicted that drug-target binding affinity in vitro would be an indicator of drug sensitivity in vivo. Although we noted a trend towards a stronger PD signal in participants with a Y79 variant, to which IMC-M113V binds with picomolar affinity, further clinical evaluation of IMC-M113V will be needed to confirm this observation. In the longer term, validated high-resolution methods and criteria may be needed to define drug sensitivity and would form the basis of a pre-screening strategy, should this be necessary to identify individuals who are most likely to benefit from treatment with IMC-M113V[51].

Multiple highly sensitive and specific assays have been developed to both qualitatively and quantitatively characterise the HIV reservoir, including methods that analyse viral nucleic acids (e.g. abundance, intactness, and integration sites), viral protein, and proviral inducibility[8,43]. These assays do not fully capture the complex interplay between the reservoir and the immune system in vivo; however, when combined with analytical ART interruption studies, they are valuable tools to inform and guide the development of HIV cure strategies. To provide a benchmark for future evaluation in multiple dose schedules, we conducted an exploratory analysis of the impact of a single dose of IMC-M113V on reservoir measures. Based on the mechanism of action of IMC-M113V, quantification of Gag p24+ cells could provide a direct assessment of on-target activity. However, the frequency of circulating p24+ cells in ART-treated individuals is below the limit of quantification when assessed ex vivo, and even with ex vivo polyclonal T cell activation, is estimated to be <10 / million CD4 + T cells, potentially necessitating the use of leukapheresis to obtain a robust signal[52]. Given the limitations on PBMC collection in this FIH study, we selected HIV CA-RNA as the assay to estimate the active reservoir size in the study participants, as viral transcription is a precursor of protein expression and importantly, this readout has also been reported as a predictor of time to viral rebound following ART cessation[53,54]. We acknowledge the limitations of HIV CA-RNA as a surrogate measure of the translationally

active reservoir due to the abundance of defective RNA transcripts[55]. We therefore also quantified intact proviruses in the participants, although it is worth noting that defective proviruses can still enable production of Gag protein[56–59]. Given the low reported expression levels of HIV proteins in infected cells, the lack of reduction of these reservoir parameters was expected and suggests that multiple dose schedules will need to include doses >15 μg, to achieve sufficient target occupancy to drive T cell-mediated reduction in the reservoir[28–30,52].

As TCR affinity enhancement may carry the risk of off-target reactivity, extensive secondary pharmacology studies were undertaken to assess this prior to clinical evaluation of IMC-M113V[35]. Importantly, we observed no off-target binding to whole blood and a panel of normal cells in vitro based on the detection of cytokines associated with CRS. The highly specific interaction of IMC-M113V with cognate pHLA complexes, coupled with its rapid plasma clearance in vivo, particularly at the low doses tested, could explain the absence of serious and/or severe adverse events, including CRS and immune effector cell-associated neurotoxicity syndrome (ICANS) in this study. ICANS was also not observed during clinical development of tebentafusp, which is cleared from the circulation within 8 h[49]. Our experience with ImmTAC and ImmTAV molecules contrasts with CD19-targeting and CAR-T cell therapies, which are frequently associated with ICANS, possibly as a result of endothelial activation and T cell trafficking to the CNS, which may increase the risk of disruption of the blood-brain barrier[46,47,60]. Finally, several skin-related TEAEs occurred in this study and were reported as possibly related, raising the question of whether these were a consequence of off-target binding. One event was considered procedure-related, and the remainder were all deemed unlikely to be related to IMC-M113V based on a lack of both a temporal relationship and biological plausibility. Furthermore, no single event occurred in more than one participant. Nevertheless, surveillance for any off-target toxicity will be paramount in the evaluation of higher doses and longer exposures.

In summary, our data demonstrate that IMC-M113V, a TCR bispecific molecule targeting HIV, has a favourable safety and tolerability profile in HLA-A*02:01-positive PLWH treated with ART for up to 7 years. Moreover, the biological activity detected after a single 15 μg dose suggests that IMC-M113V is sensitive to the levels of HIV Gag protein typically expressed during ART. These data provide initial proof of concept for this approach to targeting HIV reservoirs. Nevertheless, we acknowledge several limitations that will need to be addressed through further clinical development. First, given the very low expression of target (pHLA) in latently infected cells due to blocks in proviral transcription and translation, we anticipate that a multiple-dose schedule incorporating higher doses will be needed to provide sufficient target occupancy to drive substantial killing of infected cells and thereby achieve the multi-log reductions that are deemed necessary for long-term viral control[61]. Second, according to the 'kick and kill' paradigm, ImmTAV administration may need to be synergised with agents that specifically increase transcription and translation of intact proviruses to enable maximal targeted immune destruction[56]. Such an approach has proved challenging to date, but it will be important to re-evaluate this should suitable compounds become available. Third, if efficacy is achieved with a multiple-dose schedule, extending the application of HIV-specific ImmTAV beyond HLA-A*02:01 will be essential to enabling broad population coverage. The frequency of HLA-A*02:01 is approximately 10-28% in East and Southern Africa, where over half of all PLWH reside[37]. Because of high genetic diversity in this region, no single allele is present in at least half the population, but coverage could be increased at least 2-fold by targeting HLA-A*02:01 and another prevalent allele[37,62]. As the ImmTAX platform is modular and has been successfully adapted to target other HLA alleles, including non-classical class I alleles, this is a feasible and important goal[63–66].

## Methods

### ImmTAV molecules, pHLA complex stability and TCR binding affinity measurements

IMC-M113V[RES] was derived from the closely related ImmTAV molecule m121[27] that was modified to meet manufacturing specifications. Late in preclinical development, a single amino acid change in the TCR domain was introduced to reduce the risk of non-enzymatic post translational modification, yielding the clinical lead candidate drug, IMC-M113V. IMC-M113V and IMC-M113V[RES] show comparable target binding affinity and specificity. In vitro studies of IMC M113V[RES] described here were performed to complement data from preclinical experiments supporting regulatory submissions, all of which were performed using IMC-M113V.

To determine the stability of Gag$_{77-85}$ pHLA complexes, their response to ILT2—which recognises the α3 domain and β2-microglobulin of HLA-A—was evaluated through biolayer interferometry (Octet® RED384—Sartorius). Soluble biotinylated Gag$_{77-85}$ pHLA complexes were immobilised on a streptavidin-coated biosensor as a ligand, and binding to soluble ILT2 was then assessed at 30 min intervals over 4 h at 37 °C. The complex $t_{1/2}$ (h) was then determined by plotting the ILT2 response against time and fitting the resulting curve using non-linear regression (GraphPad Prism v8.3 or later). Binding affinities of IMC-M113V or IMC-M113V[RES] to Gag$_{77-85}$ pHLA complexes were determined by single-cycle kinetic analyses using surface plasmon resonance (SPR) (Biacore 8K) at the specified temperatures. Dissociation rate constant ($k_d$) values lower than 9.63E-06 s$^{-1}$ were constrained when determining the binding affinities ($K_D$).

### Analysis of Gag$_{77-85}$ variant prevalence

Nucleotide sequences (and metadata) were downloaded from the Los Alamos National Laboratory website (https://www.hiv.lanl.gov/ accessed 17th September 2019) and trimmed to the region containing Gag$_{77-85}$ with some additional sites on either side. The trimmed sequences were then aligned to the HXB2 reference sequence and translated to the amino acid sequence; any indels or sequences that did not provide full coverage of Gag$_{77-85}$ were excluded. Only one sample of each patient identifier was included in the analyses; sequences with no patient ID were assumed to be from unique individuals. Gag$_{77-85}$ inter-subtype variation was determined from this dataset and then combined with the published regional estimates of subtype prevalence to extrapolate per-region diversity[33].

### Cell lines and primary cells

T2 cells (#CRL-1992) were obtained from American Type Culture Collection (ATCC; Manassas, VA). C8166 cells (#88051601) were obtained from European Collection of Authenticated Cell Cultures (ECACC, UK) and genetically modified to express HLA-A*02:01 and beta-2-microglobulin using lentiviral transduction. Both cell lines were cultured in Roswell Park Memorial Institute (RPMI) medium (Thermo Fisher Scientific 42401018) supplemented with 10% foetal calf serum, 1% (v/v) penicillin/streptomycin, and 2 mM of L-glutamine. Primary Human Bronchial Epithelial Cells (HBEpC) were obtained from Promocell (#C-12640) and cultured in Promocell airway epithelial cell growth medium. IPSC derived astrocytes (Astro1g) were obtained from Cellular Dynamics (#ASC-100-020-001-PT) and cultured in iPSC Astrocyte maintenance medium. Induced Pluripotent Stem Cells (iPSC) derived cardiomyocytes were obtained from Cellular Dynamics (#C1006), NCardia (#PCMI-1031-1) and Takara Bio (Cellartis) #Y10076 and cultured according to the supplier's instructions. PBMC were obtained from Cellular Technologies Ltd (Cleveland, OH, USA), Discovery Life Sciences (Huntsville, AL, USA – previously Conversant Bio) or Stemcell Technologies GmbH (Cologne, Germany). CD8 + T cells were enriched from PBMC by positive selection using magnetic beads in accordance with the manufacturer's instructions (CD8 microbeads, 130-045-201, Miltenyi Biotech, Surrey, UK). Healthy donors gave

written informed consent to provide blood samples in compliance with an approved study protocol. Approval was obtained from the National Research Ethics Committee South Central - Oxford A Research Ethics Committee, UK A) (reference:13/SC/0226) in accordance with the principles of the Declaration of Helsinki.

## T cell redirection assays

T2 cells were incubated with Gag$_{77-85}$ peptides for 2 h before co-culture overnight with HIV-negative donor PBMC at a previously optimised effector/target ratio, together with IMC-M113V or IMC-M113V$^{RES}$ at specified concentrations. IFN-γ enzyme-linked immunospot (ELISpot) assays were performed according to the manufacturer's recommendations (BD Biosciences #551849); spots were quantified using the BD ELISpot reader (Immunospot Series 5 Analyzer, Cellular Technology Ltd, Shaker Heights, OH, USA). IL-6, IFN-γ, and TNF were quantified in the supernatants using an MSD immunoassay (V-PLEX proinflammatory kit; Meso Scale Technologies) following the manufacturer's recommendations.

## Quantification of pHLA complexes on Gag$_{77-85}$ positive cells

Epitope quantification on the cell surface was performed as follows[23,34]: HLA-A*02:01-transduced C8166 cells were infected with HIV-1 IIIB (Centre for AIDS Reagents, National Institute for Biological Standards and Control) by spinoculation. After 48 h, cells were stained with 50 nM rabbit Fc-tagged IMC-M113V$^{RES}$ (TCR domain), goat anti-rabbit-CF640R (Biotium # BT20176-1), and Alexa Fluor 488 Annexin V (Thermo Fisher Scientific # A13201), then fixed with 4% paraformaldehyde and transferred to poly-L-lysine-coated glass-bottomed chambers. Tile images (phase-contrast and annexin V) were acquired using a Nikon ECLIPSE Ti2 microscope with an oil 100 × objective to identify alive cells before Z-stack (0.8 μm apart) fluorescent images were captured covering the entire 3D cell surface. Fluorescent spots (TCR bound to pHLA = 1 epitope) on each Z-stack were counted (minimum 40 cells/condition) using the Nikon NIS-Elements software. For peptide-pulsed targets, T2 cells were incubated with Gag$_{77-85}$ peptide for 2 h, following which staining and imaging were performed as described above.

## Infected cell elimination assay

pNL4-3 (Centre for AIDS Reagents, National Institute for Biological Standards and Control, #ARP2006; donated by Dr Malcolm Martin courtesy of the NIH AIDS Research and Reference Reagent Program) plasmid was mutated to express either Gag$_{77-85}$ cognate (SLYNTVATL) or 79 F variant (SLFNTVATL) sequence. HLA-A*02:01-transduced C8166 cells were infected with HIV-1 IIIB or mutated pNL4-3 by spinoculation. After 48 h, infected cells were co-cultured with CD8 + T cells purified from HIV-negative donor PBMC at a 1:2 ratio, together with IMC-M113V at the specified concentrations for 7 days. Remaining infected cells were quantified by intracellular Gag p24 staining, which was performed along with CD3 and CD8 antibodies following live/dead staining, fixation, and permeabilisation (Supplementary Table 2)[67]. Flow cytometry was performed using the MACSQuant X (Miltenyi), and results were analysed using FlowJo (v10) using the gating strategy shown in Supplementary Fig. 7. Per cent elimination of HIV-infected cells was calculated using the following equation: [(% p24 + cells without IMC-M113V−% p24 + cells with IMC-M113V)/(% p24 + cells without IMC-M113V)] × 100. Apparent negative elimination values observed at the lower concentrations of IMC-M113V tested were normalised to zero.

## Specificity assessment against whole blood and normal cells

For whole blood cytokine release assays, fresh whole blood from healthy donors ($n = 9$) was incubated with IMC-M113V alone, IMC-M113V and exogenous Gag$_{77-85}$ peptide, or stimulatory anti-CD3 antibody (positive controls) for 4 h. Cytokine release was measured in serum supernatants and compared to negative controls (blood incubated for 4 h in the absence of IMC-M113V) using an MSD immunoassay (V-PLEX proinflammatory kit; Meso Scale Technologies, Rockville, MD). For assays involving iPSC-derived cardiomyocytes ($n = 3$ lots) or astrocytes, cells were incubated with IMC-M113V and HIV-negative donor ($n = 3$) PBMC (effector/target ratio of 10:1) for 48 h[68]. Cytokine release was measured in the supernatants using an MSD immunoassay. To assess IMC-M113V reactivity to bronchial epithelial cells ($n = 3$ lots), these cells were incubated with IMC-M113V and HIV-negative donor ($n = 3$) PBMC for 24 h, following which IFN-γ ELISpot assays were performed as described above. The assays were performed in triplicate using 96-well plates. IMC-M113V, together with the Gag$_{77-85}$ peptide, was included as a positive control.

## Study design and participants

IMC-M113V-103 (also known as Soluble T cell Receptors In Viral Eradication, STRIVE) is a phase 1/2a, open-label dose escalation study (EudraCT 2021-002008-11, 29 Oct 2021). Given the requirement for the HLA-A*02:01 allele and the restriction of prior ART experience to a maximum of 7 years for eligibility, a multi-centre design was used. The primary objective was to determine the safety and tolerability of IMC-M113V when administered as a single dose during ART, which would guide the selection of doses for evaluation in multiple dose schedules. A starting dose of 1.6 μg was selected based on the minimum anticipated biological effect level (MABEL), in accordance with European Medicines Agency guidance[69,70]. Dose escalation increments were maximally 3-fold, provided there were no emerging safety signals, and escalation was permitted by the modified toxicity probability interval (mTPI-2) algorithm[71]. Dose escalation could continue until either of the following occurred: (1) at least 50% participants (minimum $n = 4$) met pre-specified criteria for pharmacodynamic activity, namely, a ≥ 4-fold increase in serum IL-6 ≤ 24 h or a ≥ 33% reduction in absolute lymphocyte count at 24 h, both relative to baseline; (2) the dose was associated with unacceptable toxicity as defined by the mTPI-2 algorithm. The rationale for stopping escalation in the SAD study once the pre-specified pharmacodynamic activity threshold had been reached was to expedite transition to the multiple ascending dose portion of the study, in which antiviral activity was to be assessed. A minimum 4-fold increase in serum IL-6 was based on prior studies that showed a positive correlation with overall survival in patients treated with tebentafusp, which uses the same anti-CD3 mechanism of action as IMC-M113V[39].

Participants were people with HIV aged 18−65 years, with a positive test for HLA-A*02:01 as assessed by a central laboratory high-resolution (4-digit) assay, treatment with ART for a minimum of 1 year and maximum 7 years at the time of planned first dose, with pVL < 50 copies/mL, CD4 + T cell count >500 cells/μL and CD4 + T cell nadir > 200 cells/μL. Participants' sex was determined based on self-report. Males and females were eligible to participate, but as this was an FIH study, sex and gender were not considered in the design. Participants were excluded if they were a known HIV controller (pVL < 2000 copies/mL in the absence of ART for at least 12 months) or had initiated ART within 12 weeks of a confirmed diagnosis of primary HIV infection, or had active co-infection with hepatitis B or C virus.

The study was conducted in accordance with the principles of the Declaration of Helsinki and Good Clinical Practice guidelines. The protocol was approved by the Medicines and Healthcare Products Regulatory Agency, UK (CTA 36781/0011/001-0001), the Federal Agency for Medicines and Health Products, Belgium (1296758) and the Agencia Española de Medicamentos y Productos Sanitarios, Spain (RD 1090/2015). Ethical approval was obtained from East of England−Cambridge East Research Ethics Committee, UK (21/EE/0242), University of Gent Medical Ethics Committee (BC-11136), Belgium and the Drug Research Ethics Committee of the Hospital Universitario Germans Trias y Pujol, Spain. The study was registered with the European Union Drug Regulating Authorities Clinical Trials Database (EudraCT number 2021-002008-11). Participating sites were: Imperial College Healthcare NHS Trust, London, UK; Guy's & St Thomas' NHS

Foundation Trust, London, UK; Manchester University NHS Foundation Trust, UK; Chelsea and Westminster Hospital NHS Foundation Trust, UK; Kings College Hospital NHS Foundation Trust, UK; Universitair Ziekenhuis Gent, Belgium; Universitair Ziekenhuis Brussel, Belgium; Hospital Universitari Vall d'Hebron, Barcelona, Spain; Hospital Universitario Germans Trias y Pujol, Badalona, Spain; Hospital Universitario Ramon y Cajal, Madrid, Spain; Hospital Clinico San Carlos, Madrid, Spain. Participants gave informed written consent before being screened for enrolment.

Study oversight was provided by a study safety team (SST), comprising the Sponsor and Investigators. The SST was responsible for dose decision-making, including escalations and modifications for safety reasons. For each cohort, dose escalation decisions were based on the SST's review of all relevant available cumulative safety and PD data. An independent Data Monitoring Committee (DMC) comprising three clinicians with relevant expertise was also convened. The DMC provided oversight of safety considerations and recommendations regarding steps to ensure participant safety and the ethical integrity of the study.

## Procedures

IMC-M113V was given intravenously over 2 h on Day 1. Premedication with antipyretic (paracetamol 1 g orally or ibuprofen 400 mg orally, or their equivalents) and an oral non-sedating antihistamine was given to all participants within 30 min prior to the start of infusion, to minimise the incidence and/or severity of CRS and infusion-related reactions, which were anticipated based on experience with T cell engagers in oncology settings[72]. IMC-M113V infusions were administered at clinical research units at the sites. Participants were observed for a minimum of 16 h for monitoring of vital signs and, if necessary, provision of supportive care, then discharged the following day. Follow-up visits were conducted at outpatient clinics at the participating sites on Days 8, 15, 22 and 29. Clinical safety assessments included complete (Screening visit) and directed physical examinations (all other visits), vital signs measurement, 12-lead electrocardiogram in triplicate, haematology, clinical chemistry [Days 1 (pre-dose), 2, 8, 15, 22, 29], CD4+ cell count [Days 1 (pre-dose), 2, 8 and 29, using a quantitative flow cytometry panel comprising antibodies to CD3, CD16, CD56, CD45, CD8, CD14, CD19] and pVL [Days 1 (pre-dose and 8 h post-end of infusion), 2, 8 and 29; Abbott RealTime HIV-1 assay, Abbott m2000 system, detection limit−40 copies/mL]. All analyses were performed at a central laboratory (Q2 Lab Solutions) according to the manufacturer's instructions.

## Pharmacokinetic assay

Plasma concentrations of IMC-M113V were measured using a validated electrochemiluminescence immunoassay (ECLIA). This assay format was based on step-wise capture of free IMC-M113V in patient serum samples via immobilised pHLA, followed by detection with reporter-labelled recombinant human CD3.

## Pharmacodynamic activity

Serum cytokines and chemokine measurements were performed at a central laboratory (Q2 Lab Solutions) in accordance with their standard operating procedures. Serum IL-6 was measured using ECLIA from Cobas C6000 (Roche Diagnostics). Serum levels of IFN-γ, TNF, IL-2, IL-8 and IL-10 were measured using the 5-plex Meso Scale Discovery electrochemiluminescence assay; the chemokine IP-10 (CXCL10) was also measured by electrochemiluminescence. LLOQ for IL-2, IL-6, IL-8, IL-10, IFN-γ, TNF, and IP-10 were 1.78 pg/mL, 2.5 pg/mL, 1.18 pg/mL, 0.60 pg/mL, 3.52 pg/mL, 1.38 pg/mL, and 2.32 pg/mL respectively. Values below LLOQ were set at the LLOQ. For analysis of T cell activation and effector function, cryopreserved PBMC were thawed, rested for 1 h at 37 °C, followed by surface marker staining or, for intracellular cytokine staining, stimulation with PMA (Abcam #ab147465) and ionomycin (Merck #I3909) for 4 h. Brefeldin A

(BioLegend #420601) was added after 30 mins. Cells were then stained with Live/Dead UV Blue (Thermo Fisher Scientific #L23105), followed by surface markers before permeabilisation and intracellular staining for 20 min at RT. Cells were fixed in 4% PFA, washed, and acquired on the ID7000 spectral analyser (Sony). Antibodies used in this panel are described in Supplementary Table 2. Flow cytometry analysis was performed on Omiq (OMIQ.ai) with the gating strategy shown in Supplementary Figs. 4 and 5. Data was exported from Omiq and plotted using GraphPad Prism 10.5.0.

## Sequencing of *gag* from HIV CA-RNA

CD4+ cells were isolated from PBMC (CD4 Microbeads, Miltenyi Biotech #130-045-101), following which RNA was extracted using the RNeasy Mini Kit (Qiagen #74104). cDNA was synthesised and amplified (RT-PCR1) using SuperScript III One-Step RT-PCR System (Thermo Fisher Scientific #11732020) with Platinum Taq DNA Polymerase (Thermo Fisher Scientific #11304011) and SUPERase·In RNase Inhibitor (Thermo Fisher Scientific #AM2694) and the following primers[14]:

*gag*_F_RTPCR1: 5′-AAATCTCTAGCAGTGGCGCCCGAACAG-3′
*gag*_R_RTPCR1: 5′-TAACCCTGCGGGATGTGGTATTCC-3′.

The following touchdown thermal cycling conditions were used[73]: 50 °C for 30 min followed by 94 °C for 2 min and then 94 °C for 30 s, 64 °C for 30 s, and 68 °C for 3 min (3 cycles); 94 °C for 30 s, 61 °C for 30 s, and 68 °C for 3 min (3 cycles); 94 °C for 30 s, 58 °C for 30 s, and 68 °C for 3 min (3 cycles); 94 °C for 30 s, 55 °C for 30 s, and 68 °C for 3 min (41 cycles); and then 68 °C for 3 min. The total reaction volume was 13.5 μL containing a maximum of 2000 ng RNA. The amplified product from RT-PCR1 was diluted with 15 μL of nuclease-free water. Two microliters of the diluted RT-PCR1 product was next amplified (PCR2) in a 25 μL reaction volume using Platinum Taq DNA polymerase High Fidelity (Invitrogen), the same thermocycling conditions as RT-PCR1 (excluding the RT 50 °C for 30 min step) and the following nested primers[14]:

*gag*_F_PCR2: 5′-GCGCCCGAACAGGGACYTGAAARCGAAAG-3′
*gag*_R_PCR2: 5′-TATCATCTGCTCCTGTATC-3′

TapeStation 2200 (Agilent) and Qubit 3.0 (Invitrogen) were used to confirm PCR2 amplification before proceeding to library preparation using Nextera XT Library Preparation Kit (Illumina #FC-131-1096) and Nextera XT Index Kit v2 Set A (Illumina #FC-131-2001) as per the manufacturer's protocol. Amplified indexed libraries were purified using Illumina purification beads and analysed on a TapeStation 2200 and Qubit 3.0 prior to producing a 4 nM library pool. The pooled library was paired-end sequenced on a MiSeq Sequencer using a 300-cycle MiSeq Reagent Kit v2 (Illumina).

For sequence analyses, Fastq files were processed as follows: Quality and adaptor trimming were performed using Trim Galore (v0.6.2) with default parameters, specifying the amplification primers and Nextera adaptors. FastQC was used to assess read quality. Mapping was carried out against the HIV-1 HXB2 reference annotation (obtained from NCBI, November 2022; ASM310297v1 / GCA_003102975.1) using BWA INDEX and BWA MEM (v0.7.17) with PCR duplicates removed. MultiQC was used to assess mapping statistics. iVar variants (v1.3.1) were used to call variants from the aligned BAM files, and iVar consensus (v1.3.1) was used to generate consensus sequences. SAMtools Depth (v1.3) was used to compute the alignment depth along the HIV-1 genome. Sequence plotting and visualisation were performed in R using the packages muscle (v3.38.0), seqinr (v4.2.16), ggplot2 (3.3.6) and ggmsa (v1.2.3). Variants obtained in the region of interest (HIV Gag$_{77–85}$) at a frequency of >3% were reported.

## Quantification of HIV *gag* CA-RNA

Quantification of HIV *gag* CA-RNA was performed using ddPCR based on the previously published method with some modifications[74]. CD4+ cells were isolated from PBMC (CD4 Microbeads, Miltenyi Biotech #130-045-101), following which RNA was extracted using the RNeasy Mini Kit

(Qiagen #74104) and quantified using the Qubit 3.0 and a Broad Range RNA Kit (Thermo Fisher Scientific #Q10211). A minimum of two replicates each containing up to 350 ng RNA were analysed using the One-Step RT-ddPCR Advanced Kit for Probes (Bio-Rad #1864021) and the QX200 ddPCR System (Bio-Rad), following the manufacturer's protocol. The following primers and probe were used for the quantification[75]:

CS-*gag* forward primers: 5′- GACTAGCGGAGGCTAGAAGGA-GAGA-3′

CS-*gag* reverse primers: 5′- CTAATTTTCCSCCDCTTAA-TAYTGACG′-3′

CS-*gag* FAM/BHQ Probe: 5′-AT + G + GGT + GC + GAGA-3′

For some participants, primers and/or probes were customised based on the *gag* sequence determined in CA-RNA (Supplementary Table 3). Droplets were prepared using an Automated Droplet Generator (Bio-Rad) and cycled at 95 °C for 10 min; 40 cycles of (94 °C for 30 s, 59 °C for 1 min) and 98 °C for 10 min, using a ramp rate of 2 °C to improve droplet separation. Droplets were analysed on a QX200 Droplet Reader (Bio-Rad) using QXManager software (Bio-Rad, version 1.2). RNA of the human TBP gene was also quantified using the TaqMan Gene Expression VIC Assay (20×) from Invitrogen (Hs00427620_m1). For this purpose, RNA was diluted 20-fold, and 2 μL was used per reaction. HIV *gag* CA-RNA values were normalised to TBP expression and are presented as HIV CA-RNA copies/$10^6$ TBP[76,77].

## Quantification of intact proviral DNA

The cross subtype intact proviral DNA assay (CS-IPDA) was performed using the QIAcuity Four 5-plex system (Qiagen) as previously described with some modifications[55]. Briefly, CD4 + T cells were isolated from cryopreserved PBMC using the EasySep CD4 T cell isolation kit (Stemcell Technologies #17952), followed by direct lysis (DLR, Viagen #301-C) of cell pellets (-1 million cells) and XhoI restriction digestion (end volume 75 μL). Samples were analysed in triplicate with 20 μL (undiluted) input per reaction for the CS-IPDA assay that was performed without the restriction enzyme XbaI. Samples were diluted 1/100 for the RPP30 normalisation assay and analysed in triplicate (5 μL input). The same primer probe sets were used except for LTR-*gag* (Supplementary Table 3). Analysis was performed using the QIAcuity Software Suite 2.5.0.1 (Qiagen).

## Ethical approvals for the use of human material

All uses of human material have been approved for this study. The use of PBMC from healthy donor volunteers for the in vitro investigations was approved by the Oxford A Research Ethics Committee under study protocol reference 13/SC/0226. All participants provided written informed consent.

Ethical approval for the use of samples obtained from PLWH treated with IMC-M113V was obtained through the IMC-M113V-103 study (EudraCT 2021-002008-11, 29 Oct 2021), which was approved by the relevant ethics bodies at each participating site as indicated in the Study design and participants section. All participants provided written informed consent.

## Statistical analysis

In vitro cytokine release assays performed for IMC-M113V specificity assessment against whole blood and normal cells were analysed using Wilcoxon matched-pairs signed rank and Holm-Šídák multiple comparison tests (p value threshold for significance (alpha) 0.05).

*P* values for changes from baseline (pre-dose) in the HIV CA-RNA and intact proviral DNA copies were calculated using a mixed-effects model and Šídák multiple comparison test. All analyses were performed using GraphPad Prism Version 10.

## Reporting summary

Further information on research design is available in the Nature Portfolio Reporting Summary linked to this article.

## Data availability

Key elements of the IMC-M113V-103 study protocol are available at the European Union Clinical Trials Register (EudraCT 2021-002008-11). A redacted version of the IMC-M113V-103 study protocol is included in the Supplementary Information file and available at CTIS - Clinical Trials in the European Union (https://euclinicaltrials.eu). Cell-associated HIV *gag* RNA sequences generated in this study have been deposited in the BioSample database under BioProject PRJNA1372654 (https://www.ncbi.nlm.nih.gov/bioproject/?term=PRJNA1372654; https://trace.ncbi.nlm.nih.gov/Traces/study/?acc=SRP650654&o=acc_s%3Aa). Source data are provided with this paper.

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

## Acknowledgements

The authors thank all participants in the study, as well as the study teams at the participating sites, for their support of this trial and the following employees of Immunocore: M.L. McCully, D. Cuckovic, and C. Perot for assistance with manuscript preparation; D. Berman, J. Suzich and M. Dar for critical review of the manuscript. This study was funded by Immunocore Ltd with support from the Bill and Melinda Gates Foundation. Research reported in this publication was also supported by the National Institute of Allergy and Infectious Diseases under Awards # UM1AI164561 / PO1 AI178376 (MD). The content is solely the responsibility of the authors and does not necessarily represent the official views of the National Institutes of Health.

## Author contributions

The study was designed by Immunocore (study sponsor) in collaboration with the authors. L.V., J.F., B.M., A.B., Y.Y., J.W., A.D.W, Z.W., P.K.S., L.D., and S.F. contributed to the conception, design and planning of the study. M.D., H.R., J.Cl., J.C., R.H., M.H., S.M., Z.W., and P.K.S. performed assays. L.V., J.F., B.M-P., J.N., S.D.A., A.J.U., S.M.G., M.B., F.A.P., V.E., B.M., and S.F. enroled and treated patients and gathered data. M.D., A.B., H.R., A.T., J.C., R.H., M.H., S.M., Z.W., P.K.S., K.T., and L.D. analysed and interpreted data. A.B., A.D.W., Z.W., P.K.S., K.T., and L.D. drafted the manuscript. All authors critically reviewed iterations of the manuscript and approved the final draft for submission.

## Competing interests

L.V. receives research grants from J&J, ViiV Healthcare and Gilead Sciences. J.N. has received fees for educational activities and/or consultancies and/or financial support for attending conferences from AbbVie, Gilead Sciences, Janssen-Cilag, Merck Sharp & Dohme and ViiV Healthcare; S.D.A has received research grants and/or consulting fees from Gilead Sciences, GSK, MSD and ViiV Healthcare. A.J.U. has received financial support for attending a conference from Gilead Sciences. M.B. has received research grants and/or consulting fees from ViiV, Gilead, MSD, GSK, Novavax, Valneva, Cipla, Mylan, Janssen, and Roche. V.E. has received fees for educational activities and/or consultancies and/or financial support for attending conferences from Gilead Sciences, Janssen-Cilag, Merck Sharp and Dohme and ViiV Healthcare. B.M. has received consultancy fees from AELIX Therapeutics SL and AbbVie and speaker fees from Gilead, Janssen and ViiV Healthcare. M.D. was supported by the NIH MDC grant RID-HIV: UM1AI164561 and PO1 AI178376. A.B., H.R., A.T., J.C., R.H., M.H., Y.Y., J.W., S.M., A.D.W., Z.W., P.K.S., K.T., and L.D. were/are employees of Immunocore Ltd. The remaining authors declare no competing interests.

## Additional information

[1]HIV Cure Research Center, Department of Internal Medicine and Pediatrics, Ghent University, Ghent, Belgium. [2]Guy's & St Thomas' NHS Foundation Trust, London; Kings College London, London, UK. [3]Department of Infectious Disease, Imperial College London; Imperial College National Institute of Health Research Biomedical Research Centre, London, UK. [4]Infectious Diseases Department, Hospital Universitari Vall d'Hebron, Barcelona, Spain. [5]Institut de Recerca Vall d'Hebron, Barcelona, Spain. [6]Department of Internal Medicine and Infectious Diseases, Vrije Universiteit Brussel, Universitair Ziekenhuis Brussel, Brussels, Belgium. [7]Regional Infectious Diseases Unit, North Manchester General Hospital, Manchester University NHS Foundation Trust, Manchester, UK. [8]Department of Infectious Diseases, Ramón y Cajal University Hospital, Madrid; Alcalá University, IRYCIS, CIBERINFEC, Madrid, Spain. [9]Department of HIV Medicine, Chelsea and Westminster Hospital, London, UK. [10]Department of Sexual Health and HIV, King's College Hospital NHS Foundation Trust, London, UK. [11]Hospital Clinico San Carlos-IdiSSC, Ciberinfec, Universidad Complutense, Madrid, Spain. [12]Department of Infectious Diseases & Fundació Lluita contra la Sida, Institute for Health Science Research Germans Trias I Pujo (IGTP), Badalona, Spain. [13]IrsiCaixa AIDS Research Institute, Hospital Germans Trias I Pujol, Badalona, Spain. [14]CIBERINFEC, Madrid, Spain; UVic-UCC, Vic, Spain. [15]Immunocore Limited, Abingdon, UK. [16]Immunocore Limited, Gaithersburg, MD, USA. ✉e-mail: Linos.vandekerckhove@UGent.be

