## [Transparent Peer Review file · Nature Communications]

Safety and biologic activity of a bispecific T cell receptor targeting HIV Gag in males living with HIV: a first-in-human trial

Corresponding Author: Professor Linos Vandekerckhove

Version 1:

Reviewer comments:

Reviewer #1

(Remarks to the Author)

This well written article investigates the affinity of a novel bispecific TCR targeting HIV Gag and CD3 in people living with HIV. The investigators first investigate the affinity of the TCR to a range of Gag variants at position 77-85 and then assess the binding of IMC-M113V to cells transduced with HLA-2 and infected with HIV. They show high affinity to a range of variants and TCR-bound pHLA complexes on the cell surface of infected cells with a median copy number of 3/cell (range 0-14). Although significantly higher than uninfected cells, there was some background binding of 1/cell (range 0-8). There was minimal off target reactivity of IMC-M113V when cytokine production was assessed in cardiomyocytes, astrocytes or lung epithelial cells. Then in a phase 1 study in PWH with HLA02.01 alleles, they showed that IMC-M113V was safe, had minimal adverse events and there was a dose dependent increase in serum IL6, an expected pharmacodynamic effect of TCR binding. Interestingly, in participants with high affinity binding of the TCR, they showed a relationship between baseline HIV CA RNA and IL6 (but this was only 3 participants). This correlation was weaker for the 7 participants with nM variants. The manuscript is clear and well written. I only had a few minor questions and comments

1. The in vitro model looking at binding of TCR (Supp Fig S1) in productively infected cells. Have any experiments been done with latently infected cells that undergo latency reversal? This might be a more realistic model to estimate sensitivity of binding
2. The sequencing of gag in cell associated HIV RNA is a very interesting angle to the study. Given it was bulk sequencing, an assessment of quasispecies was probably not possible. Could limiting dilution sequencing of CA HIV also be performed?
3. Did the investigators look at the frequency of p24+ cells in participants? The HIV flow assay could also be helpful and may be a more direct read out of the potential effects of IMC-M113V on the reservoir.
4. In the adverse events, skin rash was quite common. What are the potential reasons for this. This should be discussed further in the discussion
5. In figure 2a, there is very wide variation in percentage elimination of HIV infected cells. Close to one log variation in the EC50. What are the potential reasons for this?
6. The HIV subtype of the participants should also be included in the patient demographics
7. The modest elevation of IL6 with the 15ug dose was encouraging. What would be a safe level of IL6 from a therapeutic such as this bispecific antibody? This should be discussed further in the discussion.
8. The potential negatives of this approach ie HLA2 specific, gag variant escape, low presentation of gag in latency and potentially toxicities, need to be better explained for a non expert audience.

Reviewer #2

(Remarks to the Author)

The submitted manuscript reports on preclinical studies and results of a first in human, single ascending dose study of a novel affinity-enhanced bispecific T cell receptor (TCR) (IMC-M113V) targeting Gag-positive cells in 12 participants, as a potential strategy for HIV elimination or functional cure. The TCR is fused to a CD3-specific chain, a variable HIV peptide Gag77-85, which incorporates the most prevalent Gag 77-85 variants. As described by the authors, the concept is adapted from TCR technology applied to cancer treatment.

The manuscript is very well written, leading to few comments or suggestions. The study design, analysis, methods and assays used are clearly and thoroughly delineated. The authors describe the design and functionality of the novel bispecific TCR targeting cells expressing Gag well.

Results presented indicate IMC-M113V was safe and well-tolerated as single doses, without cytokine release syndrome or ICANS. Notably and as described by the authors, this strategy appears to have measurable impact at low levels of HIV gag protein in vitro, which is relevant to strategies targeting the latent reservoir. Impact was suggested by levels of serum IL-6 which correlated with drug-sensitive virus in participants, assessed by target binding affinity. Results are to inform a subsequent study of multiple doses of IMC-M113V at higher doses.

Specific suggestions are listed below:

Abstract:

- No recommended edits

Introduction

- Line 65-68. Would revise to clarify that the HIV reservoir also resides in other cells in addition to CD4+ cells.
- Lines 80-82 – consider providing a reference to support the statement; two are provided in the next sentence but confirm both apply to both sentences.
- Line 88-90 – consider referring to Figure 1a here to help describe/visualize the TCR receptor.
- Line 112 – would define “PD” with first mention.
- Consider in the Intro or discussion providing information on the prevalence of individuals, or PWH if known, who are HLA-A*02:01 positive, to help frame the impact this strategy could have, or how it could be expanded/adapted to allow it to be effective for PWH without that HLA allele.

Results

- Lines 143 – 146. Consider discussing as a limitation the overlap in the median copy of complexes/cell between infected and uninfected.
- Lines 175 – 177. The numbers presented on participants prescreened in the text is n=89, but in Figure 3 it is 74. If 47 were excluded due to negative HLA-A*02:01, that leaves 27 participants (not 26). Please reconcile.

Methods

- Well-delineated

Discussion

- Authors address the limitation of using HIV cell-associated RNA as to measure/estimate the active HIV reservoir.

Figures

- Would make the font size for some legends larger as they are difficult to read, i.e. Figure 5
- Consider making the figure of the bispecific TCR more detailed and reference Figure 1a (as suggested above) to help explain the novel design.
- Figure S2 – would either describe what “index” refers to or label more specifically.

Reviewer #3

(Remarks to the Author)

Vandekerckhove and colleagues describe the evaluation of an experimental bispecific soluble TCR targeting a HIV-1 Gag77-85 peptide restricted by the prevalent HLA-A*02:01 epitope (IMC-M113V) in a first-in-human single administration, dose ranging study in 12 virally suppressed volunteers with chronic HIV-1 infection to assess the safety/tolerability and pharmacodynamics (PD) of the product as a novel approach to reduce the HIV-1 viral reservoir.

The biological plausibility of this approach was described in the Introduction as these biospecifics can target “undruggable” epitopes on T-cells as peptide and not protein recognition is sufficient; high levels of expression are not needed; intracellular peptides can be detected by cell-surface TCR detection; the product carries an scFv with an immune synapse elimination effector mechanism of action. The first in drug class biospecific ImmTAC tebentafusp is FDA approved for uveal melanoma and is being studied for reservoir reduction in chronic HBV infection as well as other pathogens. Previous in vitro studies with IMC-M113V showed promise in re-directing T-cells to eliminate HIV-1 infected cells derived from ART suppressed individuals after viral reactivation. Thus, the next step in evaluating this product as a possible drug for HIV-1 sustained viral remission was a first-in-human safety/tolerability and PD study.

Volunteers were chosen with high circulating CD4+ T-cell levels and high historical nadirs (relatively intact immune systems); viral suppression greater than 1 year; cART duration less than 7 years (maximize target availability). Study product was directly assessed for PD. Plasma IL-6 cytokine levels were assessed as a primary marker for PD activity (immune impact) as this correlation of surrogacy was established for tebentafusp. The product was well tolerated. No cytokine release syndrome episodes were noted nor were any SAEs. Dose dependent IL-6 levels were noted and highest in volunteers with the highest IMC-M113V sensitive Gag77-85 sequences in baseline cell associated RNA. The authors conclude that IMC-M113V should advance to more extensive clinical testing as an approach to achieving a functional HIV-1 cure (sustained cART-free viremic remission).

The pre-clinical data demonstrate high sensitivity of the TCR with Gag binding at picomolar concentration, as well as potency against HIV-infected cells. The clinical data from a first-in-human dose escalation study, most importantly, showed few adverse events. However, there was little evidence for effector function with only an increase in IL-6 activity, but no

changes in other cytokines. There was no change in cell-associated HIV RNA or HIV proviral DNA, although this would have been a high bar for a first-in-human study. The strength of the study lies in the platform (immune mobilizing monoclonal T cell receptor) which was previously used in cancer, to now target a virus. Decreasing the HIV reservoir is critical to achieving a functional cure of HIV, and new mechanisms will help achieve this goal.

The weaknesses of the study lie in the lack of sufficient data to demonstrate effector function. The authors use IL-6 as the marker of T-cell activity but it's unclear why other cytokines such as IFN- γ aren't elevated. Assessing other markers of T-cell activation such as CD69 positivity would have also been helpful. Overall, the authors present a novel protein for HIV elimination using a platform that has been proven in cancer treatment, but with limited exploration of T-cell activation in humans.

Attention to the points below would strengthen an already strong manuscript.

Major points:

1. A brief comparison to other CD3-HIV co-binding proteins would be helpful here (e.g., the HIV env/CD3/CD28 tri-specific antibody in Promsote et al., Nat Comm 2023).
2. The authors include a brief discussion of bNAbs for HIV treatment, particularly for the viral reservoir. It should be noted that some studies have shown reductions in viral reservoir, particularly when started in acute HIV (e.g., Gunst et al, Nat Med 2022).
3. Figure 2c: The choice of cytokines presented here and throughout the paper should be better justified. Here the authors use IFN- γ as the marker of activity, but throughout the rest of the paper, present IL-6 as the primary marker. This is critical because of the varying responses between IFN- γ and IL-6 (both elevated in vitro in Figure 2c, but variable response in humans in Figure S3). Please provide justification for using IFN- γ in the off-target reactivity study.
4. Figure 2c: The figure suggests that IMC-M113V induces less IL-6 than the anti-CD3 alone but induces similar IFN- γ to anti-CD3 alone. This raises the question if this due to greater CD4+ T-cell activation in the anti-CD3 alone group. Please clarify if these are statistically significant differences, and if so, why that might be the case.
5. Figure 2c and lines 168-172: The authors state that "No cytokine production above background levels was observed in whole blood..." This statement should be supported by a statistical test. Visually, Figure 2c shows increasing IL-6 and IFN- γ production at increasing concentrations of IMC-M113V even in the absence of Gag peptide, at least in whole blood.
6. Figure S3 and lines 280-283: Why were there few changes in other cytokines? The prior study by Middleton et al. on tebentafusp is used to justify IL-6 as a primary correlate of activation, but that study also found increased IL-10 which was not increased here. Perhaps this is due to the biomass of target in a solid tumor model vs the HIV-1 viral reservoir overall, but this should be noted by the authors here. Furthermore, the preclinical data (Figure 2c) showed dramatic increases in IFN- γ which we again don't see here, which begs the question of the degree of T-cell activation. Did the authors look at markers of T-cell activation such as CD69?
7. Lines 322-324: It would be helpful to address the limitation of the HLA restriction. The HLA-A*02:01 restriction in uveal melanoma does not present a large barrier to utility; for HIV, it severely limits the impact. This study is certainly an excellent proof-of-concept, but there is little overlap globally between the specific HLA subtype and HIV prevalence (Olivier, JAMA Net Open 2023).
8. Better define the strengths and weakness of measuring impact on the viral reservoir. While the authors are appropriately circumspect about making much of a safety and PD study, they do employ an intact proviral DNA assay (IPDA) as an indication of reservoir impact. I would strongly encourage this team to collaborate with labs that are skilled in reservoir measurements (Siliciano, Trautmann, Deeks, Douek, Chaumont, etc.). For the next phase of clinical evaluation, the team would probably want to use culture-based assays such as quantitative viral outgrowth assays (QVOA) which are as close to the gold standard now in 2025. There are others such as TILDA, etc.
9. Strongly consider using scRNASeq for next studies to better define surrogates of activity.
10. Taken together, I would urge the authors to note that these are future directions as the clinical development plan matures.

Minor points:

1. Figure 1d: it is surprising that half of the uninfected cells had >1 pHLA-TCR complex. This is only slightly lower than a median of 3 in infected cells. but I wonder how prevalent the Gag peptide used here is in uninfected cells.
2. Table 1 and Figure 5: It is worth noting that one patient was previously a long-term non-progressor, and that same patient also had an actively decreasing CD4 count (lower at enrollment than prior nadir by 100 cells/mL). Furthermore, this patient also had the greatest fold increase in IL-6. Is this a coincidence or is there something unique in this patient's T-cell response?
3. Discussion: The authors state, "We anticipate that a multiple dose schedule incorporating higher doses will be needed." This certainly appears to be true, and as such, but it is best to not overstate the lack of off-target binding. First, it is not certain that there is no off-target binding without statistical tests in Figure 2c, and even so, higher doses may result in more off-target binding.
4. Methods: The authors should justify the use of pre-treatment with an antipyretic and antihistamine for all volunteers and note this limitation when assessing the adverse events data.
5. Figure 5c and S3: The authors can consider presenting these figures with absolute values rather than fold-change, which can be difficult to interpret given the LLOQ.
6. Define the acronym PD as pharmacodynamics at first mention.

(Remarks to the Author)

The authors present preclinical results and an open-label single dose escalation study of a first in-human TCR bispecific molecule targeting HIV Gag epitope 77-85 in HLA-A*02+ individuals on ART. The treatment was administered to 12 participants and follow-up was conducted at various timepoints for 29 days. The data suggests the molecule does not have off-target effects and clears cells that present the epitope by HLA-A*02. The treatment is well-tolerated, and the authors evaluated the pharmacodynamics of the agent in vivo. The findings show there was no impact on the reservoir/proviral load using the CS-IPDA assay, even transiently, in any patient. Overall, the manuscript is clear, concise and well written. Major and minor comments for the authors are highlighted below:

MAJOR:

1. It is unclear whether, in the authors' opinion, there is any signal of efficacy to justify an ongoing study testing this agent at higher dose levels. While it is appreciated that the safety signal is high, without any signal of efficacy it is unclear how dose escalating with the increased toxicity expected, is justified. This should be discussed along with the potential reasons (other than dose) why no signal of efficacy was observed?
2. While this approach has been used with success in melanoma, the approach in HIV is yet to be realised. Further, it is telling that the majority of participants enrolled on this study are white which is the population where HLA A0201 is the most prevalent. Can the authors speculate on the broader impact of this HLA A0201-restricted therapy to the PLWH population given the prevalence of the virus in the African continent? This also warrants discussion.

MINOR:

- Please confirm that all participants were high resolution typed at A0201. And if not, please justify why. In the abstract, the authors should specify primary and secondary endpoints assessed.
- Lines 125-128: The IMC-M113VRES molecule should be more clearly introduced. Authors should note Y79F is the most prevalent variant when it is first mentioned (which is stated later in line 158).
- Lines 129-135: Authors should specify how many variants were identified and how many they tested.
- Lines 153-156: These lines should be rewritten for clarity as it was hard to follow.
- Line 154: C8166 is introduced here as an HIV-infected CD4+ T cell line, but this should be mentioned sooner in line 142.
- Line 176: The participant numbers in this line differ from the CONSORT diagram in Fig. 3. This line states eligible participants is n=89, however Fig. 3 shows n=74. This line states the HLA-A*02+ participants is n=33, however Fig. 3 shows n=26.
- Line 177: The authors should specify the screening criteria that resulted in 4/16 failing the screen. Additionally, these criteria should be added to Fig. 3, as exclusion criteria is listed in the figure too.
- Line 244-246: These lines should be rewritten to improve flow and clarity, and specify the pM and nM affinity ranges.
- Line 246: The authors should specify the pM affinity variant.
- The IMC-M113VRES molecule was assessed often in the Results section but was not discussed in the Discussion.
- Line 433-435: Equations should be referenced in the main text.
- Line 466: This states the criteria threshold is 50% participants, but states the minimum is n=4; should it be n=6?
- Line 519-520: Amend Supplementary Table 2 to add all fluorophores listed in these lines.
- Some other limitations should be addressed in the discussion: n=1 in Cohorts 1 and 2, and the short window of follow-up post-infusion.
- The journal expects that the title and/or abstract must indicate when findings apply to only one sex or gender.
- The methods section should include whether sex and/or gender were considered in the study design.
- Add n values (# participants or replicates) to all figure legends where applicable.
- For the present study, the ART regimens reported in Table 1 do not seem necessary to reveal.
- CONSORT checklist not coinciding with correct page numbers in manuscript.

Reviewer #5

(Remarks to the Author)

Version 2:

Reviewer comments:

Reviewer #1

(Remarks to the Author)

The authors have adequately addressed all my questions and concerns

Reviewer #2

(Remarks to the Author)

The authors have effectively addressed my comments in the rebuttal and revised manuscript. The authors have also effectively addressed the comments and concerns of the other reviewers. I do not have any additional comments or concerns and feel the revisions have improved the manuscript significantly. I support publication of the manuscript.

Reviewer #3

(Remarks to the Author)

The authors have provided highly detailed and effective responses to all of the myriad of questions that were raised to the critique. The process was a lovely scientific dialogue, as well, and not merely a punch list of answers to get the paper across the finish line. Kudos.

Reviewer #4

(Remarks to the Author)

The authors have addressed all my questions.

Reviewer #5

(Remarks to the Author)

Response to Referees

We would like to thank the reviewers for their constructive comments and suggestions.

Reviewer 1

Although significantly higher than uninfected cells, there was some background binding of 1/cell (range 0-8).

The background binding seen with the uninfected cells (HIV pHLA-negative) in Figure 1d can be attributed to the non-specific binding of the goat anti-rabbit*CF640R secondary antibody. The infected C8166 cells stained with only the secondary antibody (no TCR) show the same 0-8 pHLA/cell range seen on the uninfected cells stained with TCR + secondary antibody.

- For clarity, we have amended the figure 1d legend to explain the ‘no TCR’ control.

The manuscript is clear and well written. I only had a few minor questions and comments.

1. The in vitro model looking at binding of TCR (Supp Fig S1) in productively infected cells. Have any experiments been done with latently infected cells that undergo latency reversal? This might be a more realistic model to estimate sensitivity of binding

We agree that this is a more realistic model. Work is ongoing to improve sensitivity of the imaging method sufficiently to detect HIV peptides on ex vivo infected cells.

- We also respectfully refer the reviewer to our prior publication: **Elimination of Latently HIV-infected Cells from Antiretroviral Therapy-suppressed Subjects by Engineered Immune-mobilizing T-cell Receptors**. Yang H et al. *Molecular Therapy* 2016, doi: 10.1038/mt.2016.114.

This publication contains comprehensive data indicating that the TCR binds to latently infected cells that have undergone latency reversal. We showed specific elimination of ex vivo CD4+ T cells from ART-treated PLWH, after activation in vitro to reverse latency. CD4+ T cells were co-cultured with autologous or allogenic CD8+ T cells together with a precursor molecule that is almost identical to IMC-M113V (same specificity and affinity). Dose-dependent elimination was observed, indicating sensitivity of TCR binding to infected cells after latency reversal.

2. The sequencing of gag in cell associated HIV RNA is a very interesting angle to the study. Given it was bulk sequencing, an assessment of quasispecies was probably not possible. Could limiting dilution sequencing of CA HIV also be performed?

Bulk sequencing of the active reservoir was performed to identify the dominant target sequence in participants in an attempt to explore potential associations with pharmacodynamic responses. This method successfully identified low frequency target variants in some of the participants (Supplementary Fig 5) but has limitations, which we acknowledge.

Limiting dilution sequencing of HIV CA-RNA is an excellent suggestion, however, it was unfortunately not possible to do this for this single ascending dose (SAD) study due to sample limitations.

In the longer term, the impact of quasispecies on the efficacy of IMC-M113V will be an important area of investigation. Limiting dilution methods, along with other approaches such as near full length sequencing, will certainly be powerful approaches to support this.

- We have amended the Discussion to address the reviewer’s point (lines 351-363).

3. Did the investigators look at the frequency of p24+ cells in participants? The HIV flow assay could also be helpful and may be a more direct read out of the potential effects of IMC-M113V on the reservoir.

1. We agree that a protein-based assay would provide a more direct read out of IMC-M113V effects on the reservoir and is therefore highly desirable. We appreciate the suggestion to do the HIV flow assay; however, we would like to highlight that Pardons et al. employed leukapheresis, which we infer was to ensure that the cell input in the assay was sufficient to yield a reproducible signal because the frequency of p24+ cells was so low (doi: 10.1371/journal.ppat.1007619). Specifically, a signal was detected in ART-treated individuals only after activation of CD4+ T cells with PMA/ionomycin and the median (IQR) frequency was 4.3 (0.7-8) per million cells.

Similarly, a FISH-Flow assay was successfully used by Gunst J et al. to demonstrate small but statistically significant reductions in the proportion of p24+ cells in participants in the eCLEAR study (<https://doi.org/10.1038/s41591-022-02023-7>). However, these participants were ART-naïve at baseline and thus had a measurable baseline signal (10-100-fold higher than in ART-treated individuals by HIV flow, Pardons et al.) that enabled detection of a post-intervention reduction.

We have not implemented the HIV flow or FISH-flow assay yet, as the majority of the current trial sites did not have access to leukapheresis, but these assays are under consideration for future studies.

- We have amended the Discussion to provide an explanation for why a p24+ assay was not used in this study (lines 372-376)

4. In the adverse events, skin rash was quite common. What are the potential reasons for this. This should be discussed further in the discussion

The majority of adverse events affecting skin (n = 5/6) were observed in one participant. The skin redness was procedure-related (a reaction to adhesive on the ECG electrodes); the herpes simplex episode occurred 5 days after study drug administration in a participant who had a prior history of HSV infection and was therefore deemed a recurrent episode. Three events (eczema and itch) occurred 20 days after administration of IMC-M113V and were therefore deemed unlikely to be related, based on the lack of temporal association, the molecule half-life of <18 hours and the lack of biological plausibility.

An episode of phlebitis occurred in another participant 16 days after IMC-M113V administration and was recorded as possibly related to IMC-M113V by the investigator. However, the Sponsor's determination was that this was also unlikely to be related, for the same reasons given for the events described above.

- We have clarified this in the Results section (lines 200-204) and have added a statement to the Discussion summarising these details (lines 400-405).

5. In figure 2a, there is very wide variation in percentage elimination of HIV infected cells. Close to one log variation in the EC50. What are the potential reasons for this?

The percent elimination in this assay indicates that IMC-M113V is highly potent in vitro (EC₅₀ values 1.2-13.1 pM). We would like to highlight that despite a ~1 log variation in EC₅₀, the values reflect extremely low drug concentrations (equivalent to ~0.1-1.0 ng/ml), which can accentuate any expected inter-donor variability. The inter-donor variability likely relates to variation in the proportions in CD8+ T cell subsets across the donors (particularly effector memory cells, as reported in Yang H et al. 2016).

6. The HIV subtype of the participants should also be included in the patient demographics

- We have added this information to Table 1.

Please note that this was obtained from viral resistance tests performed at clinical sites as part of standard of care and a viral subtype was not available for all participants.

7. The modest elevation of IL6 with the 15ug dose was encouraging. What would be a safe level of IL6 from a therapeutic such as this bispecific antibody? This should be discussed further in the discussion.

Thank you for the question. In our opinion, a safe threshold cannot be defined because: 1) while IL-6 is a key mediator of toxicity in CRS, the pathology is driven by a cascade of immune activation events, with

other cytokines and various immune cell populations contributing to organ dysfunction; 2) patient-specific factors can increase the risk for CRS of Grade ≥ 2 (hypotension or hypoxia), eg. lung disease, adrenal insufficiency; 3) our oncology studies show that there is overlap in the range of IL-6 levels across severity grades.

In case of interest, the mean (SEM) level of serum IL-6 at 8 hours post-dose in patients receiving tebentafusp for metastatic cutaneous (mCM) or uveal (mUM) melanoma is shown below (Hamid, O. et al. Tumor microenvironment (TME) features and serum cytokines in patients with metastatic uveal and cutaneous melanoma treated with tebentafusp. EJC Skin Cancer, Volume 2, 100140).

	Serum IL-6 (pg/ml) - mean (SEM)	
	mCM	mUM
No CRS	24.1 (7.7)	20.3 (12.8)
Grade 1	30.7 (9.2)	433.7 (159.5)
Grade 2	126.8 (55.2)	1267 (377.5)
Grade ≥ 3	N/A	34071.6 (32543.5)

These data provide some insight into the relationship between serum levels and CRS severity grade. However, for the reasons outlined above, avoidance of CRS depends on careful monitoring for clinical symptoms and signs, particularly at the first dose, and serum cytokine levels are confirmatory rather than predictive.

The duration of the cytokine response is also pertinent. The IL-6 elevations in this study were transient, (consistent with the serum half-life of IMC-M113V), therefore, it is expected any cytokine-related adverse events – even at doses $>15 \mu\text{g}$ - would be transient, as has been noted with tebentafusp in patients with uveal melanoma (<https://doi.org/10.1158/1078-0432.CCR-25-1513>).

- We have included a statement in the Results on the highest IL-6 value recorded in this study (lines 244-245) and modified the text in the Discussion to link the lack of CRS to the low levels of cytokines observed (lines 315-316). As CRS could also be the result of off-target toxicity, we have added text to highlight the importance of monitoring for this when evaluating higher doses and multiple dose schedules (lines 405-406).

8. The potential negatives of this approach ie HLA-A2 specific, gag variant escape, low presentation of gag in latency and potentially toxicities, need to be better explained for a non expert audience.

We acknowledge these potential limitations of the approach and have expanded the discussion to explain how these have been or will be addressed through clinical development.

- Considerations for expanding to other HLA alleles beyond HLA-A2 are addressed in lines 422-430.
- Overcoming and / or mitigating Gag variant escape and low presentation of Gag in latency are addressed in lines 288-306 and lines 413-422.
- Toxicities and mitigation strategies are discussed in lines 319-324 and lines 388-406.

Reviewer 2

Comments on the Introduction:

- Line 65-68. Would revise to clarify that the HIV reservoir also resides in other cells in addition to CD4+ cells.

- We have amended this sentence as suggested (now lines 64-66).

- Lines 80-82 – consider providing a reference to support the statement; two are provided in the next sentence but confirm both apply to both sentences.

- We have added references to this sentence (now lines 78-84).

- Line 88-90 – consider referring to Figure 1a here to help describe/visualize the TCR receptor.

- Thank you for the suggestion – now included.

- Line 112 – would define “PD” with first mention

- This has been corrected (now line 114).

*Consider in the Intro or discussion providing information on the prevalence of individuals, or PWH if known, who are HLA-A*02:01 positive, to help frame the impact this strategy could have, or how it could be expanded/adapted to allow it to be effective for PWH without that HLA allele.*

Thank you for the suggestion.

- We have clarified the prevalence of HLA-A*02:01 in the Results section (lines 179-181) and expanded the Discussion to explain the estimated prevalence of HLA-A*02:01 in PLWH, including estimates based on recent high-resolution HLA typing in selected populations in East and Southern Africa (lines 424-425).
- We have added a statement to explain that the ImmTAX platform can be deployed to produce TCR bispecific molecules that target other HLA alleles, with references to support this (lines 428-430).

Results: Lines 143 – 146. Consider discussing as a limitation the overlap in the median copy of complexes/cell between infected and uninfected.

Per our response to Reviewer 1, there is some background binding with the uninfected cells (HIV pHLA-negative), which can be attributed to the non-specific binding of the goat anti-rabbit*CF640R secondary antibody, as indicated by the same pHLA/cell range seen in the ‘no TCR’ control.

- We have amended the text in the Results (lines 149-151) to reflect this point.

In addition, some overlap in the signal from infected and uninfected experimental conditions could also reflect the presence of p24-low cells in the former (Fig. 1c shows that the MFI of p24 expression in infected cells spans a 1-2 log range), together with a range in the level of HLA class I on the cell surface. The reported range in pHLA copies/cell cannot be further refined, which is a limitation of this method.

*Results: Lines 175 – 177. The numbers presented on participants prescreened in the text is n=89, but in Figure 3 it is 74. If 47 were excluded due to negative HLA-A*02:01, that leaves 27 participants (not 26). Please reconcile.*

Thank you for pointing out this error. The text is correct and Figure 3 has been corrected.

Discussion

- Authors address the limitation of using HIV cell-associated RNA as to measure/estimate the active HIV reservoir.

- This is discussed in lines 380-382.

Figures

- Would make the font size for some legends larger as they are difficult to read, i.e. Figure 5

- We have increased the font size in the graphs in all figures.

- Consider making the figure of the bispecific TCR more detailed and reference Figure 1a (as suggested above) to help explain the novel design.

- We have amended Figure 1a and included more details to explain the design and mechanism. We hope these changes have helped to make it clearer.

Figure S2 – would either describe what “index” refers to or label more specifically.

- We have changed ‘index’ to ‘cognate’ and amended the figure legend to explain what this means.

Reviewer 3

General: The weaknesses of the study lie in the lack of sufficient data to demonstrate effector function. The authors use IL-6 as the marker of T-cell activity but it's unclear why other cytokines such as IFN- γ aren't elevated. Assessing other markers of T-cell activation such as CD69 positivity would have also been helpful. Overall, the authors present a novel protein for HIV elimination using a platform that has been proven in cancer treatment, but with limited exploration of T-cell activation in humans.

This SAD study achieved its primary objective by identifying the 15 μ g dose as a suitable initial step dose to test in a subsequent multiple ascending dose (MAD) study. Demonstration of T cell effector function was an exploratory objective that will be assessed in depth in the ongoing MAD study. Nevertheless, we appreciate the concern regarding the lack of impact on cytokines other than IL-6 and the lack of analysis of effector function beyond IL-6. We have addressed this in our response to comments #3 and #6 below.

1. A brief comparison to other CD3-HIV co-binding proteins would be helpful here (e.g., the HIV env/CD3/CD28 tri-specific antibody in Promsote et al., Nat Comm 2023).

We appreciate the suggestion. The HIV trispecific antibody, N6/ α CD3- α CD28, binds to HIV Env. Promsote et al. reported that this antibody elicited high levels of CD4⁺ and CD8⁺ T cell activation in the presence of HIV-infected cells in vitro, with evidence of some activation in the absence of HIV. The molecule also elicited HIV antigen expression and T cell activation in ex vivo assays with PBMC from ART-treated individuals. In an in vivo study, naïve Rhesus macaques developed high levels of T cell activation after IV or SC administration of the antibody, together with broad induction of proinflammatory chemokines and cytokines to substantial levels that were, surprisingly, not associated with adverse events. However, a similar trispecific antibody, α CD38/ α CD3- α CD28 (SAR442257), was tested in a Ph1 trial in refractory / relapsed multiple myeloma and non-Hodgkin's lymphoma. The study was terminated during dose escalation due to safety events (CRS in over half the participants, recurrence of CRS, and high rates of CMV and EBV reactivation). <https://doi.org/10.1182/blood-2024-204614>

This study highlights the risk of combined CD3/CD28 agonism, even with monovalent antibodies, and in our opinion, confirms that the high levels of T cell activation in the NHP study were concerning, despite the apparent lack of adverse effects.

Two other T cell engager approaches are briefly summarized below, in case of interest to the reviewer. As there are no clinical publications on these we have not discussed them in our manuscript.

- HIV dual-affinity retargeting agents (DARTs) comprise Env-CD3 co-binding proteins and were shown to specifically engage HIV-infected target cells in vitro and re-direct T cell killing of these targets in vitro and ex vivo (reviewed by Nordstrom et al. <https://doi.org/10.1016/j.jve.2022.100083>). In SHIV-infected macaques on ART, the DART molecules bound to circulating T cells without causing T cell activation or cytokine induction. No impact on viral parameters was observed. A first-in-human study evaluating single and multiple dose schedules of one of the HIV DARTs (MGD014) in PLWH reported that the molecule did not induce activation markers or serum cytokines despite binding to the majority of CD4⁺ and CD8⁺ T cells in the blood (Norstrom J et al., IAS 2022, abstract # OAA0403).
- Sengupta et al (doi: 10.1073/pnas.2123406119) reported on the in vitro evaluation of a CD3 bispecific incorporating a TCR mimic that binds an HIV Pol peptide in complex with HLA-A*02:01 as the targeting arm. The molecule was potent against infected primary CD4⁺ T cells in vitro, but to the best of our knowledge no clinical trials have been initiated.

2. The authors include a brief discussion of bNAbs for HIV treatment, particularly for the viral reservoir. It should be noted that some studies have shown reductions in viral reservoir, particularly when started in acute HIV (e.g., Gunst et al, Nat Med 2022).

- We have noted this and have amended the text in the introduction to include reference to Gunst et al. 2022 (lines 78-84). The study by Gaebler et al., which showed a statistically significant reduction in the intact intact reservoir, is also cited (lines 82-84 and reference #16).

We note that Gunst et al reported a reduction in intact proviruses in the eCLEAR study, which assessed bNAbs +/- HDAC inhibitor in PLWH at ART initiation versus ART alone. In this well-designed trial, they also reported a significant decrease in intact proviruses from baseline through one year in the ART-only arm. Of interest, there was also a reduction in the active reservoir (proportion of p24+ cells) in two of the intervention arms. Because eCLEAR enrolled a mix of people with recent (<6 months) and chronic infection it is difficult to ascertain the relative contributions that infection recency and administration of bNAbs at the time of ART initiation – a unique aspect of this study – made to the reported outcomes.

We are also aware of the data from the RIO and FRESH studies that showed a positive effect of bNAbs on post-treatment control in PLWH who initiated ART during acute infection. To our knowledge these studies have not yet been published.

3. Figure 2c: The choice of cytokines presented here and throughout the paper should be better justified. Here the authors use IFN- γ as the marker of activity, but throughout the rest of the paper, present IL-6 as the primary marker. This is critical because of the varying responses between IFN- γ and IL-6 (both elevated in vitro in Figure 2c, but variable response in humans in Figure S3). Please provide justification for using IFN- γ in the off-target reactivity study.

- We have amended the Results (lines 170-172 and 224-232) to provide a clearer explanation of the rationale for the choice of read-outs for the preclinical and clinical data that are presented in the manuscript.

In brief, in this manuscript a variety of read-outs were used to evaluate IMC-M113V preclinically and clinically. IFN- γ is a sensitive and direct measure of T cell activation in vitro and was therefore used for initial screening for potency (on-target activity) of IMC-M113V (Figure 1, Supp Figure S1b) and off-target reactivity in assays with a panel of normal cells (Figure 2c). A granzyme B Elispot assay was also performed in parallel; this was not included in this manuscript because the results were similar to those obtained with the IFN- γ assay. For the whole blood assay, we used a panel of cytokines - IFN- γ , TNF- α , IL-6, IL-2 and IL-1 β - but included only the IFN- γ and IL-6 data in Figure 2c, as similar results were obtained with the other cytokines.

- We can include the data for the other cytokines evaluated in the preclinical assays if the reviewer considers this to be essential.

For the clinical study we evaluated changes in multiple serum cytokines as a secondary endpoint. (A copy of the study protocol is provided with this submission). A 4-fold increase in serum IL-6 was selected as a criterion of biological (pharmacodynamic) activity to inform dose escalation decisions. This choice of cytokine was based on clinical data with tebentafusp, which showed substantially greater fold changes in serum IL-6 than IFN- γ in patients with metastatic uveal melanoma (mean ~35-fold increase for IL-6 vs. 7-fold for IFN- γ (Middleton M et al. 2020, doi: 10.1158/1078-0432.CCR-20-1247 and Carvajal R et al. 2022, doi: 10.1200/JCO.21.01805). The same phenomenon was observed in this SAD study.

- This is explained in lines 224-228 and discussed in lines 325-330.

4. Figure 2c: The figure suggests that IMC-M113V induces less IL-6 than the anti-CD3 alone but induces similar IFN- γ to anti-CD3 alone. This raises the question if this due to greater CD4+ T-cell activation in

the anti-CD3 alone group. Please clarify if these are statistically significant differences, and if so, why that might be the case.

This is an intriguing observation. We have not investigated this because the experimental condition in question is the Gag₇₇₋₈₅ peptide ('pep.') positive control, in which target cells are saturated with peptide (10 μ M) and cultured with PBMC effectors plus a very high concentration (5 nM) of IMC-M113V. We wish to emphasise that the anti-CD3 antibody condition is included only as a second positive control and there was no intent to make comparisons between these two controls.

5. Figure 2c and lines 168-172: The authors state that "No cytokine production above background levels was observed in whole blood..." This statement should be supported by a statistical test. Visually, Figure 2c shows increasing IL-6 and IFN- γ production at increasing concentrations of IMC-M113V even in the absence of Gag peptide, at least in whole blood.

- Thank you for the recommendation. We have included a statistical test and amended the Results text (lines 172-175) and figure legend.

We would like to highlight that there was no significant increase in IL-6 in the whole blood assay, nor IFN- γ in the normal cell screen in the presence of IMC-M113V. There was a significant increase in IFN- γ in the whole blood assay at the 10 nM concentration only. As this concentration is outside the anticipated clinical dose range (based on the range of drug concentrations at which there is no broad loss of specificity against a panel of normal cell types), this result is not considered clinically relevant. We have included a statement to clarify this in the Results (lines 175-176).

6. Figure S3 and lines 280-283: Why were there few changes in other cytokines? The prior study by Middleton et al. on tebentafusp is used to justify IL-6 as a primary correlate of activation, but that study also found increased IL-10 which was not increased here. Perhaps this is due to the biomass of target in a solid tumor model vs the HIV-1 viral reservoir overall, but this should be noted by the authors here. Furthermore, the preclinical data (Figure 2c) showed dramatic increases in IFN- γ which we again don't see here, which begs the question of the degree of T-cell activation.

We agree with the comment that the size of the active HIV-1 reservoir (i.e. the fraction that expresses viral antigens) is likely to represent a small biomass of target relative to that of a solid tumour, and is therefore a key factor determining the magnitude of all cytokine responses in the study participants. This also explains the apparent disconnect between the IFN- γ signal in vitro and in the clinical study.

- We have amended the Discussion to make the point that the target burden (reservoir size) and low dose is one of the likely explanations for the minimal changes in IFN- γ and other cytokines including IL-10 (lines 316-319).

In Figure 2c, the strong IFN- γ signal from the positive control reflects both high target expression (cells pulsed with a high concentration of peptide) and high IMC-M113V concentration (5 nM), both of which do not reflect conditions in PLWH (intentionally, given that this is a positive control for the cross-reactivity assay).

The other key determinant is dose. This is borne out by emerging data from our ongoing MAD study

- We refer the reviewer to data showing dose-dependent cytokine elevations (presented by Mothe B et al. at CROI 2025, abstract #104; <https://www.croiconference.org/abstract/1861-2025/>) which are **appended to this document, together with additional confidential data.**

These data show that a consistent increase in IFN- γ is observed only at doses $\geq 300 \mu$ g, and typically is only ~2-fold relative to baseline, whereas at lower target doses (60 and 120 μ g), only 1/5 participants showed an increase of similar magnitude. In this SAD study, only 1/10 participants in the 15 μ g dose cohort showed a >2-fold increase above pre-dose levels (Supplementary Fig. 3), which is consistent with the trend observed in the ongoing MAD study.

The lack of IL-10 induction in this SAD study (vs. Middleton 2020) is consistent with the lack of induction of IFN- γ in the majority of participants, since IL-10 typically follows IFN- γ in acute inflammatory responses (as shown by Stacey A et al, 2009; <https://doi.org/10.1128/jvi.01844-08>).

The known sensitivity of IL-6 as an indirect biomarker of T cell activation was the justification for its use in guiding dose escalation decisions (as explained in our response to Comment #3 above). The observed response rate of a >4-fold increase in 5/10 participants in the 15 μ g cohort in this SAD study bears this out.

- We address the apparent disconnect between clinical IL-6 and IFN- γ responses (and other cytokines) in the Discussion. We include a reference to a study of blinatumomab (CD3 \times CD19 bispecific) in minimal residual B-lineage acute lymphoblastic leukemia (MRD+ B-ALL), a low tumour burden setting, where elevations in IL-6 without a concomitant IFN- γ were observed in several patients (lines 342-350).

Did the authors look at markers of T-cell activation such as CD69?

- We appreciate the question. To address the reviewer's concern, we have included new data in the manuscript (**shown below and now included in Figure 5**) providing evidence for CD4+ and CD8+ T cell activation and upregulation of cytolytic molecules post-dosing.

This was performed in a subset of participants from the 15 μ g dose cohort, based on sample availability. The Results (lines 249-257), Discussion (lines 339-344) and Methods (lines 625-635) have been updated and the gating strategy is included as a new supplementary figure (Figure S4). In the Discussion we make the point that the changes in CD69 expression are commensurate with the low dose and anticipated low target expression. We believe this is a reasonable conclusion given that no change in T cell activation markers was observed in PLWH in the Phase 1 study of MGD014, the Env-specific DART, discussed in point #1 above.

7. Lines 322-324: It would be helpful to address the limitation of the HLA restriction. The HLA-A*02:01 restriction in uveal melanoma does not present a large barrier to utility; for HIV, it severely limits the impact. This study is certainly an excellent proof-of-concept, but there is little overlap globally between the specific HLA subtype and HIV prevalence (Olivier, JAMA Net Open 2023).

Thank you for this comment and for mentioning the Olivier et al. study, which we agree makes a critical point regarding the risk of racial and ethnic inequities in trial eligibility for HLA-based therapeutics and the need to find solutions. Specifically, the development of HLA-based therapeutics for HIV must address differences in HLA allele frequencies in the African continent vs. EU/US.

Regarding the comment that the HLA restriction severely limits impact, this was also raised by Reviewers 1 and 2. However, allele frequencies alone do not provide a complete picture and consideration of the absolute numbers of potentially eligible patients is important. We outline these below, basing our estimations on recently published high resolution HLA typing in Eastern and Southern African populations (Banjoko et al, 2025. <https://doi.org/10.1038/s41598-025-06704-4>).

- The HLA-A*02:01 allele frequency in this study was 5% in S. Africans, vs. 11% in African Americans and 22% in European Americans (broadly similar to Olivier et al). These frequencies translate to HLA-A*02:01 percentages in the respective populations of ~9.75%, 21% and 39%.
- Considering that there are an estimated 7.7 million adults living with HIV in S. Africa, a potential ~0.75 million PLWH could be eligible for treatment with IMC-M113V based on HLA-A*02:01 prevalence. This is greater than the estimated number of HLA-A*02:01-positive PLWH in EU5 and USA combined (0.65 million, adjusted for demographic mix in these countries).

Banjoko et al. also show that HLA-A*02:01 is among the top 5% most frequent alleles in Eastern and Southern African populations, therefore, adequate population coverage will likely require targeting HLA-A*02:01 alongside other prevalent alleles in these regions.

- We have expanded the Discussion to discuss how the restriction of the current drug to HLA-A*02:01 alone can be overcome (lines 422-430).

8. Better define the strengths and weakness of measuring impact on the viral reservoir. While the authors are appropriately circumspect about making much of a safety and PD study, they do employ an intact proviral DNA assay (IPDA) as an indication of reservoir impact. I would strongly encourage this team to collaborate with labs that are skilled in reservoir measurements (Siliciano, Trautmann, Deeks, Douek, Chaumont, etc.). For the next phase of clinical evaluation, the team would probably want to use culture-based assays such as quantitative viral outgrowth assays (QVOA) which are as close to the gold standard now in 2025. There are others such as TILDA, etc.

- We have expanded the Discussion to better address the strengths and weaknesses of measuring impact on the viral reservoir (lines 364-384).

We greatly value the experience and expertise of labs skilled in reservoir measurements and will continue to liaise with experts in this area. We appreciate the advice on the alternative reservoir assays (measuring the replication competent reservoir and the frequency of cells with msRNA) and will aspire to expand on the analyses described here in future studies.

9. Strongly consider using scRNASeq for next studies to better define surrogates of activity.

We appreciate the suggestion. We are intending to incorporate this in the ongoing MAD study.

10. Taken together, I would urge the authors to note that these are future directions as the clinical development plan matures.

Thank you for your constructive comments and suggestions.

Minor comments

1. Figure 1d: it is surprising that half of the uninfected cells had >1 pHLA-TCR complex. This is only slightly lower than a median of 3 in infected cells. but I wonder how prevalent the Gag peptide used here is in uninfected cells.

The Gag peptide is not present in uninfected cells. The background binding seen with the uninfected cells (HIV pHLA-negative) can be attributed to the non-specific binding of the goat anti-rabbit*CF640R secondary antibody. The infected C8166 cells stained with only the secondary antibody (no TCR) show the same 0-8 pHLA/cell range as the uninfected cells stained with TCR + secondary antibody.

- We have amended the text in the Results (lines 149-151) to clarify this point.

As clarified in our response to Reviewer 2, the low median pHLA copy/cell in the infected cell condition and the overlap in this signal with uninfected cells could also reflect the presence of p24-low cells in the former (Fig. 1c shows that the MFI of p24 expression in infected cells spans a 1-2 log range), together with a range in the level of HLA class I expression on the cell surface. The reported range in pHLA copies/cell cannot be further refined, which is a limitation of this method.

2. Table 1 and Figure 5: It is worth noting that one patient was previously a long-term non-progressor, and that same patient also had an actively decreasing CD4 count (lower at enrollment than prior nadir by 100 cells/mL). Furthermore, this patient also had the greatest fold increase in IL-6. Is this a coincidence or is there something unique in this patient's T-cell response?

The CD4 nadir in this participant was >4 years prior, before he started ART. Values for Screening, Week 1 and Week 2 were 760, 474 and 613 cells/ μ L (all within normal range). This degree of fluctuation is very

common, irrespective of HIV status, and does not indicate an actively decreasing CD4 count. It likely reflects fluctuations in absolute lymphocyte count since the CD4% values for the same time points were 40.2, 42.7, 39.6% (also all within normal range). We state in the Results that two participants (including this one) had a CD4 count <500 cells/ μ L on Day 1 despite normal values at Screening (lines 207-209).

- We have amended the footnote in Table 1 to clarify that the count was normal at Screening.

It is intriguing that this individual had the greatest fold increase in IL-6. Given that they had lost their long-term non-progressor status several years prior to enrolling in the study, we have avoided speculating on the relevance of any distinct immunological features of this phenotype. However, identifying baseline covariates that may predict response to treatment is an important goal for ongoing and future studies.

3. Discussion: The authors state, "We anticipate that a multiple dose schedule incorporating higher doses will be needed." This certainly appears to be true, and as such, but it is best to not overstate the lack of off-target binding. First, it is not certain that there is no off-target binding without statistical tests in Figure 2c, and even so, higher doses may result in more off-target binding.

Thank you for this comment. As stated in our response to your major comment #5, we have included a statistical test in the text and figure legend and amended the Results text. This supports our contention that there was no off-target binding in vitro.

- The text now reads: No statistically significant cytokine production above background levels was observed in whole blood, or in co-cultures of immune cells and cardiomyocytes, astrocytes or lung epithelial cells, when incubated with IMC-M113V at concentrations below 10 nM. As the biologically active dosing range was anticipated to be 0.1-1 nM, the reactivity at 10 nM was not considered to be a concern.

We agree that the risk of off-target binding may be increased at higher doses, however, the extensive normal cell screening that was performed as part of the regulatory submission demonstrated a lack of off-target binding at a wide range of clinically relevant drug concentrations. Further details can be provided if needed.

- To address the reviewer's concern, we have amended the Discussion to emphasise the importance of monitoring for safety signals that may be caused by off-target binding, irrespective of preclinical safety data, throughout clinical development (lines 388-406).

4. Methods: The authors should justify the use of pre-treatment with an antipyretic and antihistamine for all volunteers and note this limitation when assessing the adverse events data.

- We have modified the text in the Methods (lines 594-598) and added a reference to a review by Geraud A et al., 2024, (<https://doi.org/10.1016/j.ejca.2024.114075>) which covers the toxicities associated with T cell engagers and their management and prevention, including use of premedication.

Regarding the suggestion that non-steroidal premedication could limit the assessment of adverse events data, while they are often used with T cell engagers to reduce the incidence of CRS or infusion-related reactions or the severity of these events once triggered, they are unlikely to mask them completely. This would be a legitimate concern with corticosteroid premedication.

5. Figure 5c and S3: The authors can consider presenting these figures with absolute values rather than fold-change, which can be difficult to interpret given the LLOQ.

We have replaced Figure 5c and Supplementary Figure S3 with new graphs showing the absolute values of IL-6 and the other cytokines analysed.

6. Define the acronym PD as pharmacodynamics at first mention.

- This has been corrected.

Reviewer 4

1. It is unclear whether, in the authors' opinion, there is any signal of efficacy to justify an ongoing study testing this agent at higher dose levels. While it is appreciated that the safety signal is high, without any signal of efficacy it is unclear how dose escalating with the increased toxicity expected, is justified. This should be discussed along with the potential reasons (other than dose) why no signal of efficacy was observed?

We appreciate the comment. We wish to clarify here - and we state in the manuscript - that the aim of this SAD study was to identify tolerable, biologically active doses for further evaluation in multiple dose schedules. The SAD design is not intended to assess efficacy but is an accepted approach to identifying a safe dose range that can be evaluated for efficacy in subsequent multiple dose schedules. This approach is consistent with EMA/CHMP guidance on risk mitigation in first-in-human trials.

https://www.ema.europa.eu/en/documents/scientific-guideline/guideline-strategies-identify-and-mitigate-risks-first-human-and-early-clinical-trials-investigational-medicinal-products-revision-1_en.pdf

As the reviewer points out, dose escalation for any new investigational agent poses a risk of increased toxicity but is necessary to identify a safe and effective dose. The study protocol (included in this submission) incorporates multiple risk mitigations for both the SAD and MAD studies to ensure that the benefit / risk ratio is favourable.

While a signal of efficacy is not a pre-requisite for exploring higher doses, we did observe a dose-dependent serum IL-6 signal (Figure 5b) which is a positive finding that provides evidence of pharmacodynamic activity. This, together with the tolerability of the doses tested to date, justified proceeding to the MAD study.

Per our response to Reviewer 3, we believe that the principal reasons for the minimal increase in serum cytokines (a possible surrogate for efficacy) other than IL-6 are the low dose range tested in this SAD study and the low biomass of target (active HIV reservoir). This is borne out by emerging data from our ongoing MAD study.

- Data showing dose-dependent cytokine elevations (Mothe B et al. CROI 2025, abstract #104 and confidential data) are **appended to this document** to illustrate this point.
- We have included new data to provide evidence of T cell activation and enhanced effector function, as requested by Reviewer 3. The **enclosed figure provided for Reviewer 3 above** (now Figure 5d in the manuscript) shows CD4+ and CD8+ T cell activation and upregulation of cytolytic molecules post-dosing.
- Please refer to our response to Reviewer 3's major comment #6 for full details.

2. While this approach has been used with success in melanoma, the approach in HIV is yet to be realised. Further, it is telling that the majority of participants enrolled on this study are white which is the population where HLA A0201 is the most prevalent. Can the authors speculate on the broader impact of this HLA A0201-restricted therapy to the PLWH population given the prevalence of the virus in the African continent? This also warrents discussion.

Thank you. This point was made by the other reviewers and we agree that further drug development targeting other HLA alleles, in addition to HLA-A*02:01, will be needed to have maximum impact in high HIV prevalence regions.

- Please refer to our response to Reviewer 3's major comment #7 and amendments to the Discussion to address this point (lines 422-430).

Regarding the majority of white participants in this SAD study, this is not wholly attributable to the HLA-A*02:01 restriction but reflects the demographics of the PLWH at the study sites, together with multiple factors that influence enrolment in a first-in-human study. For example, women from African countries

who are of child-bearing potential account for the majority of WLWH at the participating site; protocol restrictions, including contraception requirements, are known barriers to participation for this group. Nevertheless, development of TCR bispecific molecules for other HLA alleles would certainly be warranted if a reproducible signal of post-treatment control is observed in the MAD study.

Minor comments *Please confirm that all participants were high resolution typed at A0201. And if not, please justify why.*

- High resolution typing was performed in all participants. This is now clarified in Methods (line 559)
In the abstract, the authors should specify primary and secondary endpoints assessed.

- We have amended the abstract accordingly.

- *Lines 125-128: The IMC-M113VRES molecule should be more clearly introduced.*

- Thank you for this comment. We have edited this section to provide more clarity (lines 125-130 and 434-441).

Authors should note Y79F is the most prevalent variant when it is first mentioned (which is stated later in line 158).

- We have now introduced Y79F as the most prevalent variant at first mention earlier in this section (lines 132-133).

- *Lines 129-135: Authors should specify how many variants were identified and how many they tested.*

- 8 variants were identified with an estimated global frequency of $\geq 2\%$ and all 8 were tested. We have amended the text to reflect this (line 135).

- *Lines 153-156: These lines should be rewritten for clarity as it was hard to follow.*

- Thank you. We have amended this text and hope it is clearer (now lines 157-160).

- *Line 154: C8166 is introduced here as an HIV-infected CD4+ T cell line, but this should be mentioned sooner in line 142.*

- Thank you. This has been corrected (now lines 145-146).

- *Line 176: The participant numbers in this line differ from the CONSORT diagram in Fig. 3. This line states eligible participants is n=89, however Fig. 3 shows n=74. This line states the HLA-A*02+ participants is n=33, however Fig. 3 shows n=26.*

- Thank you for pointing out this error. We have corrected Figure 3.

- *Line 177: The authors should specify the screening criteria that resulted in 4/16 failing the screen. Additionally, these criteria should be added to Fig. 3, as exclusion criteria is listed in the figure too.*

- We have updated the text and Figure 3 to clarify the reasons for screen failure.

- *Line 244-246: These lines should be rewritten to improve flow and clarity, and specify the pM and nM affinity ranges.*

- Thank you for the suggestion. We have modified the text to address this and included the affinity ranges (now lines 270-274).

- *Line 246: The authors should specify the pM affinity variant.*

- We have included the details.

- *The IMC-M113VRES molecule was assessed often in the Results section but was not discussed in the Discussion.*

- We appreciate that this may have caused confusion. We have added text to further clarify the use of IMC-M113VRES in the Results and Methods, as pointed out earlier. Because it has the same specificity as the clinical molecule, IMC-M113V (1 amino acid difference in one of the CDRs) and it was used to complement preclinical assays that supported the regulatory submission, which were performed only with nominated clinical candidate drug IMC-M113V, we think it is more appropriate to explain this in the Methods rather than the Discussion (lines 434-441).
- *Line 433-435: Equations should be referenced in the main text.*
- We have included a reference in the Results section where potency against HIV-infected cells in vitro is presented (lines 160-161).
- *Line 466: This states the criteria threshold is 50% participants, but states the minimum is n=4; should it be n=6?*

The criteria allowed dose escalation until pre-specified criteria for pharmacodynamic activity were met in a minimum of 2/4 participants. This criterion was met in the 15 µg cohort after the first 4 participants had received IMC-M113V, so dose escalation was halted, and we expanded the cohort to a total of 10 participants to provide a more robust evaluation of this dose (as permitted by the protocol).

- This point is explained in the Results (lines 193-196).
- *Line 519-520: Amend Supplementary Table 2 to add all fluorophores listed in these lines.*
- We have updated Supplementary Table S2 to include this information.
- *Some other limitations should be addressed in the discussion: n=1 in Cohorts 1 and 2, and the short window of follow-up post-infusion.*

We appreciate that the rationale for single patient cohorts for Cohorts 1 and 2 was not made explicit in the text. This strategy is not a limitation but an important aspect of the study design, which contributes to a favourable benefit / risk ratio by minimizing exposure of participants to sub-therapeutic doses.

- We have added explanatory text in the Results (line 193) and included a reference to a publication explaining the utility of accelerated titration designs for Phase 1 studies.

We do not consider the window of 28 days' follow-up to be a limitation, given that:

- 1) this was a single dose study and the observed PK profile of IMC-M113V indicates that it has a very short serum half-life;
- 2) toxicity associated with ImmTAX molecules is primarily within the first hours-days of first exposure to a given dose; cumulative toxicity is not observed. This is highlighted in a recent publication on tebentafusp that showed that most treatment-related adverse events occurred within the first 4 weeks, during administration of step-up doses (treatment in this study was given once weekly until disease progression occurred. DOI: 10.1056/NEJMoa2304753).

- *The journal expects that the title and/or abstract must indicate when findings apply to only one sex or gender.*

- We have amended the title accordingly.
- *The methods section should include whether sex and/or gender were considered in the study design.*
- The relevant section in the Methods has been updated accordingly (lines 562-564).

Add n values (# participants or replicates) to all figure legends where applicable.

- We have included n values in all figures and / or figure legends.
- *For the present study, the ART regimens reported in Table 1 do not seem necessary to reveal.*

- We appreciate this thoughtful comment and have removed this information.
- *CONSORT checklist not coinciding with correct page numbers in manuscript.*
- We have updated the CONSORT checklist to ensure reference to correct page numbers.

Changes in serum cytokines in the IMC-M113V multiple ascending dose study (ongoing)

The above data from Cohorts 1-3 (15/30/60, n = 5; 20/40/120, n = 5; 20/40/300 mcg, n = 6) were presented at CROI 2025 (Mothe B et al., abstract #104). Confidential data from Cohort 4 (20/40/300/600 mcg) from the ongoing MAD study is included on the right. Enrolment to this cohort is ongoing. S1, S2, S3 refer to step doses. T1 refers to the target dose for each cohort (60, 120, 300 and 600 mcg respectively).

To aid interpretation of the above data, we show the maximum fold change (FC) for each participant and each cytokine below.